# BuildingsBench: A Large-Scale Dataset of 900K Buildings and Benchmark for Short-Term Load Forecasting

**Patrick Emami, Abhijeet Sahu, Peter Graf**
National Renewable Energy Lab
{Patrick.Emami, Abhijeet.Sahu, Peter.Graf}@nrel.gov

## Abstract

Short-term forecasting of residential and commercial building energy consumption is widely used in power systems and continues to grow in importance. Data-driven short-term load forecasting (STLF), although promising, has suffered from a lack of open, large-scale datasets with high building diversity. This has hindered exploring the pretrain-then-fine-tune paradigm for STLF. To help address this, we present BuildingsBench, which consists of: 1) Buildings-900K, a large-scale dataset of 900K simulated buildings representing the U.S. building stock; and 2) an evaluation platform with over 1,900 real residential and commercial buildings from 7 open datasets. BuildingsBench benchmarks two under-explored tasks: zero-shot STLF, where a pretrained model is evaluated on unseen buildings without fine-tuning, and transfer learning, where a pretrained model is fine-tuned on a target building. The main finding of our benchmark analysis is that synthetically pretrained models generalize surprisingly well to real commercial buildings. An exploration of the effect of increasing dataset size and diversity on zero-shot commercial building performance reveals a power-law with diminishing returns. We also show that fine-tuning pretrained models on real commercial and residential buildings improves performance for a majority of target buildings. We hope that BuildingsBench encourages and facilitates future research on generalizable STLF. All datasets and code can be accessed from https://github.com/NREL/BuildingsBench.

## 1 Introduction

Residential and commercial buildings in the United States are responsible for about 40% of energy consumption and 35% of greenhouse gas emissions [10]. Globally, these estimates are respectively 30% and 27% [22]. Building energy demand forecasting plays a part in reducing global emissions by helping to decarbonize the building sector.

Short-term load forecasting (STLF), which typically ranges from hour-ahead to a few days ahead, plays a multitude of critical roles. STLF can help match shifting energy supplies with customer demand, as well as aid energy markets with accurately setting prices based on forecasted supply/demand [19]. Accurate forecasts can be directly used by reinforcement learning [44] and model predictive control [1, 9] for optimal building energy management.

However, STLF remains a challenging problem, as energy demand can fluctuate heavily due to a variety of unobserved and exogenous factors. As such, data-driven methods have risen in popularity to address STLF [2, 52]. Interest in these techniques has also been spurred by a rise in deployments of advanced sensors (i.e., smart meters) that record building energy consumption. However, the number of instrumented buildings with publicly released data remains low. Moreover, it is typical for multiple years of historical data to be reserved for training and validation, with trained models tested

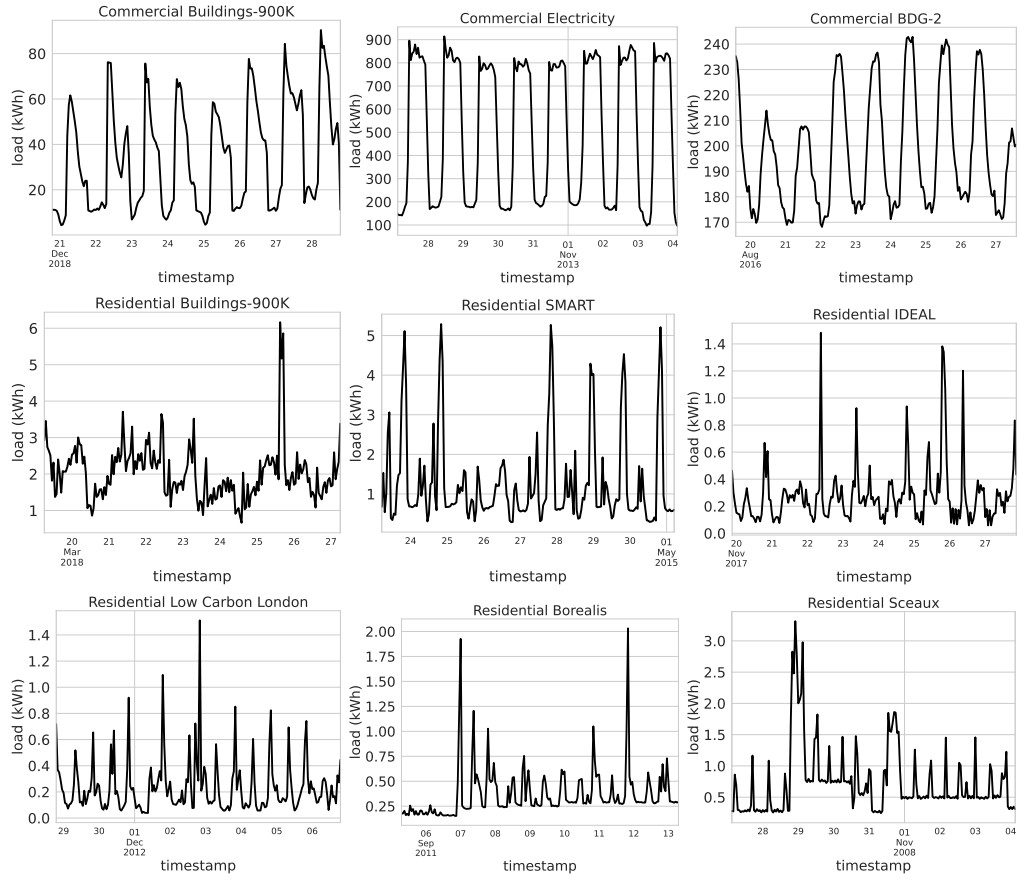

Figure 1: **BuildingsBench gallery**. Top row: commercial buildings (farthest left is simulated commercial Buildings-900K data). Second and third rows are residential buildings (farthest left in the second row is simulated residential Buildings-900K data).

on a single held-out year *for the same building* [28]. This incurs a lengthy data collection period per building that is not scalable. There is thus a lack of sufficiently large publicly available datasets (Table 1), which has hindered the investigation of large-scale pretraining (i.e., one model trained on every building [7, 17]).

In this work, we introduce BuildingsBench, an open-source platform for large-scale pretraining and for benchmarking zero-shot STLF [7, 17] and transfer learning for STLF [48, 14]. In zero-shot STLF, models forecast loads for unseen target buildings *without any fine-tuning*. This drastically reduces time-to-deployment on newly instrumented buildings. In transfer learning, a pretrained model is fine-tuned on a target building, assuming a limited yet realistic amount of data (e.g., 6 months).

As part of BuildingsBench, we introduce Buildings-900K, a dataset of nearly one million *simulated* time series, which approaches the scale of datasets in natural language processing and computer vision. This data is sourced from the NREL End-Use Load Profiles (EULP) database [45]. The EULP is a multiyear, multi-institution effort to create a statistically representative database of the entire U.S. building stock's energy consumption via careful calibration and validation of physics-based building simulations. We also provide an evaluation suite that combines 7 open datasets totalling over 1,900 *real* buildings (Table 2). Example time series from the simulated and real datasets are shown in Figure 1. Compared to other existing datasets (Table 1, Table 2), our large-scale pretraining and evaluation data contains both simulated and real building energy consumption time series spanning a wider range of geographic locations, years, and types of both residential *and* commercial buildings.

Our platform automates the evaluation of a variety of simple and advanced baselines on the two tasks. Our results on zero-shot STLF reveal that synthetically pretrained models achieve strong performance on real commercial buildings. We observe a smaller distribution shift between simulated

and real commercial buildings than residential buildings. We also show that pretrained models can further improve performance by fine-tuning on real commercial and residential building data. Buildings-900K also enables studying large-scale pretraining of transformers on geographical time series. The utility of transformers for forecasting has recently been questioned [49], but a lack of sufficiently large public time series datasets has made investigating this challenging. Our main finding is a power-law scaling with diminishing returns between dataset scale (size and diversity) and generalization (for commercial buildings). We expect that BuildingsBench will facilitate research on new modeling techniques and large-scale datasets for generalizable STLF.

To summarize, we contribute: 1) Buildings-900K, a simulated dataset for large-scale pretraining, 2) a platform for benchmarking zero-shot STLF and transfer learning, and 3) valuable insights on model pretraining for STLF.

## 2    Short-Term Load Forecasting

The BuildingsBench benchmark considers the following univariate forecasting problem. Given $H$ past load values $x_{t-H:t}$ and $H + T$ covariates $z_{t-H:t+T}$ (defined in Section 3.1), we aim to predict the conditional distribution for $T$ unobserved future load values $y_{t+1:t+T}$:

$$p(y_{t+1}, \ldots, y_{t+T} \mid x_{t-H}, \ldots, x_t, z_{t-H}, \ldots, z_{t+T}). \tag{1}$$

We consider a day-ahead STLF scenario with $H = 168$ hours (one week) and $T = 24$ hours. A primary goal of BuildingsBench is to study and evaluate STLF models, which learn a single set of parameters $\theta$ shared by all buildings for the distribution in Eq. 1. We use a probabilistic formulation for our benchmark, as applications of STLF increasingly require uncertainty estimates, such as planning and scheduling of renewable energy sources for buildings [19].

## 3    The Buildings-900K Dataset

In this section, we introduce our dataset for pretraining STLF models.

**Simulated data source:** Our dataset is sourced from the NREL EULP database [45]. The EULP provides 15-minute resolution appliance-level consumption simulated with EnergyPlus [8, 29] for a base set of 900K building models (550K residential and 350K commercial) spread across all climate regions in the U.S. It aims to provide a statistically representative picture of the entire U.S. buildings stock at various aggregation levels (county, state, etc.) and under various electrification scenarios. The building simulations were extensively calibrated over a three-year

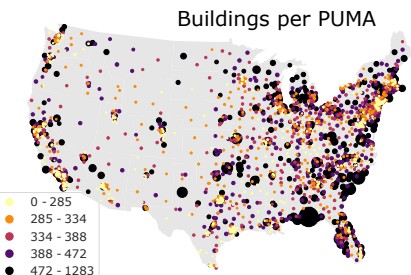

period with advanced metering data ($\sim 2.3$ million meters), data from 11 utilities, as well as other public/private datasets related to energy usage. Socio-economic building characteristics, including location based on Public Use Microdata Area (PUMA, $\sim$2400 PUMAs in the U.S.—see inset), are sampled from distributions generated from U.S. Census survey responses. Please see Wilson et al. [45] for the complete description or App. B.3 for more details.

**Processing and storage:** To create Buildings-900K, we extract 900K total energy consumption time series (in kilowatt-hours (kWh)) from each of the non-upgraded buildings in the 2021 version of the EULP. To promote accessibility of our dataset, we also aggregate the 15-minute resolution to hourly to reduce the size. This data requires about 110 GB to store, significantly less than the entire EULP (70+ TB). We store all buildings within each PUMA in a single Parquet file by year (there are two years, 2018 and an aggregated "typical meteorological year" (TMY) [45]) and by building type (residential/commercial), which amounts to 9,600 Parquet files. Each file has a column for the timestamp and a column for each building's energy consumption (8,760 rows per file). Processing the EULP to extract this data took $\sim$3 days with Apache PySpark on a 96-core AWS cloud instance.

**Pretraining splits and loading:** A validation set is created by withholding the final two weeks of 2018. The test set consists of buildings in four PUMAs that are withheld from both training and validation. All splits use a 24-hour sliding window to extract 192-hour load sub-sequences. Because shuffling large datasets of sub-sequences is computationally demanding, our platform provides a

Table 1: Comparing popular building energy consumption datasets to our dataset Buildings-900K. Our dataset has significantly more buildings (residential and commercial) located across a wide geographic area and spans multiple years, which enables studying large-scale pretraining for STLF.

| | # buildings | Open access | Residential | Commercial | Total hours | # Sites |
|---|---|---|---|---|---|---|
| Pecan Street [40] | 1,000 | ✗ | ✓ | ✗ | ? | ? |
| Electricity [41] | 370 | ✓ | ✗ | ✓ | ~12.9M | 1 |
| Building D.G.P. 2 [27] | 1,636 | ✓ | ✗ | ✓ | ~53.6M | 19 |
| Low Carbon London [30] | 5,567 | ✓ | ✓ | ✗ | ~97.5M | 1 |
| Buildings-900K (*simulated*) | **900,000** | ✓ | ✓ | ✓ | **~15B** | **2400** |

Table 2: BuildingsBench STLF evaluation data.

| | Residential | Commercial | Years spanned | # Sites | Total days |
|---|---|---|---|---|---|
| Buildings-900K-Test (*simulated*) | 915 | 565 | TMY, 2018 | 4 | 1.1M |
| BuildingsBench (*real*) | 953 | 970 | 2007–2018 | 10 | 1.2M |

custom PyTorch [33] `Dataset` that creates an index file to map a shuffled list of integers to a subsequence. Each line of the index file is accessible in $O(1)$ time with Python's `seek` function. Apache PyArrow is used to efficiently load the indexed building's time series into memory.

**Hosting and licensing:** Buildings-900K is hosted by the Open Energy Data Initiative and is available for download with a permissive `CC-4.0` license (link available in App. A).

## 3.1 Feature Extraction

Beyond the load time series, we also extract these covariates on-the-fly to condition forecasts on:

- Calendar features—day of the week, day of the year, and hour of the day—are generated from the timestamp, which we then cyclically encode with sine and cosine.
- The latitude and longitude coordinate of the centroid of the building's PUMA (using metadata provided by the EULP) for encoding geospatial information.
- A binary feature for building type (residential (0) or commercial (1)).

## 4 BuildingsBench Evaluation Platform

BuildingsBench provides an open-source software platform for evaluating models on zero-shot and transfer learning for STLF using a collection of *real building datasets*. To avoid confusion, in our analysis (Sec. 5), we will use the name BuildingsBench to refer *only to the real building evaluation data* and explicitly state whether the results are for Buildings-900K-Test (*simulated*) or for BuildingsBench (*real*) (Table 2). We now describe each task in more detail.

**Zero-shot STLF:** For this task, pretrained models are asked to provide day-ahead forecasts for unseen simulated and real buildings. Our use of "zero-shot" here refers to a setting where only one week of historical data is presumed available, and is thus insufficient for training or fine-tuning a deep neural network. All available years are used for each test building.

**Transfer learning**: Our transfer learning task assumes 6 months of metered data has been collected for each target building, about ~180 training samples. Fine-tuned models are then tasked with day-ahead forecasting for the next 6 months using a 24-hour sliding window.

## 4.1 Real Building Datasets

Here, we briefly describe the real building data used for evaluating the two tasks and defer additional details to App. C. Altogether, our real building benchmark has over 1,900 buildings and 1.2M days of energy usage.

**Electricity** [41]: 370 commercial buildings in Portugal with data spanning 2011–2014.

**Building Data Genome (BDG) Project 2** [27]: 1,636 commercial buildings across 19 sites in 2016 and 2017. We include buildings from 4 U.S. sites (Panther, Fox, Bear, Rat).

**Low Carbon London** [30]: Energy consumption meter readings from 5,567 London, UK households between 2011–2014. To keep the overall number of residential and commercial buildings roughly the same, we keep a random sample of 713 buildings.

**SMART** [5]: Meter readings for 7 homes in western Massachusetts, U.S., between 2014–2016.

**IDEAL** [34]: Electricity meter data from 255 homes in Edinburgh, UK, between 2016–2018.

**Individual household electric power consumption** [15]: Energy consumption from a single household in **Sceaux**, Paris, between 2007–2010.

**Borealis** [4]: 6-second load measurements recorded for 30 homes in Waterloo, ON, in 2011–2012.

**Processing and storage:** We resample each time series to hourly if the data is sub-hourly. Smart meter time series typically have missing values, sometimes extending over weeks or months. Buildings with more than 10% of the data missing were not included. For included buildings, we linearly interpolate missing values, and if values are missing for a span greater than 1 week, we fill this with zeros. We also provide an option to exclude buildings with a max hourly consumption > 5.1 MW (only 15 from electricity) to keep the range of consumption values similar between Buildings-900K and BuildingsBench test data, as extrapolation is not our focus. A small fraction of spikes in the time series due to noisy meter readings are classified as outliers and filtered with a non-parametric distance-based sliding window algorithm [25] (see App. C.8). Each annual consumption time series per building is stored as a CSV file.

**Hosting and licensing:** We host the processed versions of each dataset, along with the original permissive licenses, alongside Buildings-900K under the same `CC-4.0` license. Our code is open-sourced under a `BSD-3` license.

## 4.2 Evaluation Metrics

We primarily evaluate task performance with two metrics, the normalized root mean square error (NRMSE) and the ranked probability score (RPS). The NRMSE (also known as the coefficient of variation of the RMSE) is widely used, as it captures the ability to predict the correct load shape. For a target building with $M$ days of load time series,

$$NRMSE := 100 \times \frac{1}{\bar{y}} \sqrt{\frac{1}{24M} \sum_{j=1, i=1}^{M, 24} (y_{i,j} - \hat{y}_{i,j})^2}, \tag{2}$$

where $\hat{y}$ is the predicted load, $y$ is the actual load, and $\bar{y}$ is the average actual load over all $M$ days. In the appendix, we also report the normalized mean absolute error and normalized mean bias error (see descriptions in App. D).

The RPS is a well-known metric for uncertainty quantification in probabilistic forecasting [13]. It compares the squared error between two cumulative distribution functions (CDFs), the predicted CDF and the observation represented as a CDF. Define the indicator function $\mathbf{1}_{x \leq y}$ as 1 if the actual load $x$ is $\leq y$ and 0 otherwise. The continuous RPS for a predicted CDF $\hat{F}$ for the load at hour $i$ is:

$$RPS := \int_0^\infty (\hat{F}_i(y) - \mathbf{1}_{y_i \leq y})^2 dy. \tag{3}$$

Our platform implements the closed-form RPS for Gaussian CDFs, as well as a discrete RPS for categorical distributions (used by baselines that discretize the positive real line for token-based load forecasting, see Sec. 4.3). These are formally defined in App. D.

## 4.3 Baselines

Various baselines are implemented and benchmarked in the BuildingsBench platform. For zero-shot STLF, we pretrain a representative time series transformer [47] on Buildings-900K. Transformers have recently gained significant interest for STLF [17, 21, 36, 50]. The model is described in brief here and with more detail in App. E.

**Transformer (Gaussian)**: This model is the original encoder-decoder transformer [43] repurposed for autoregressive time series forecasting, as proposed in Wu et al. [47]. It predicts a Gaussian distribution at each time step $t + i$ conditioned on the past week and the previous $i - 1$ predictions. We train it to minimize the Gaussian negative log-likelihood loss. Because our demand time series are highly non-Gaussian (periods of low consumption interspersed with bursts), the loss diverges during training when we use global standard scaling to normalize the data. The Box-Cox power transformation [6] removes this instability. To compute the RPS, we approximate a Gaussian in the unscaled space by backprojecting the scaled standard deviation (see App. D for details).

**Transformer (Tokens)**: We also implement a transformer variant that uses *tokenization* to predict discrete load tokens. This baseline allows us to test whether quantizing loads into tokens is beneficial for large-scale pretraining. We also found that a comparison between tokenization and Gaussian time series transformers was missing from the literature. The model is trained to predict a categorical distribution over a vocabulary of load tokens by minimizing a multi-class cross-entropy loss. To quantize loads, we use a simple strategy—`faiss-gpu` [23] KMeans clustering with $K = 8{,}192$ fit to the Buildings-900K training set. We merge clusters $<10$ Watts apart to obtain a compact vocabulary size of 3,747 tokens. See App. E for analysis on our tokenizer design.

Each transformer is pretrained jointly on residential and commercial buildings at three different sizes: **Transformer-S** (3M params), **Transformer-M** (17M params), and **Transformer-L** (160M params). Models are trained on 1 billion load hours with early stopping. See App. E for more architecture details, App. F for hyperparameter tuning details (for learning rate and batch size), and App. G for training compute requirements.

We also benchmark **persistence** forecasting on both the zero-shot STLF and transfer learning tasks:

**Previous Day:** For scenarios where the load conditions change relatively slowly, the previous day's load is a strong baseline for day-ahead STLF [42].

**Previous Week:** The 24-hour load profile on the same day from the previous week is used [17].

**Ensemble:** This persistence baseline computes a Gaussian distribution for each predicted hour $t + i$ whose mean is the average load at hour $t + i$ over the past 7 days:

$$\hat{\mu} = \frac{1}{7} \sum_{j=1}^{7} x_{(t+i-24j)}; \quad p(y_{t+i}|x_{t-H:t}) := \mathcal{N}\left(\hat{\mu}, \sqrt{\frac{1}{7} \sum_{j=1}^{7} (x_{(t+i-24j)} - \hat{\mu})^2}\right). \quad (4)$$

In STLF, outperforming persistence is an indicator that a model is producing meaningful forecasts. On the transfer learning task, we also benchmark the following forecasting baselines by training them directly on the target building's 6 months of meter data with supervised learning:

**LightGBM:** The light gradient-boosting machine (LightGBM) [24] is a popular decision-tree-based algorithm suitable for STLF [32]. We use the multi-step forecasting implementation from `skforecast` [3] with 100 estimators and no max depth.

**Linear regression, DLinear:** Inspired by Zeng et al. [49], we benchmark a linear *direct* multi-step forecaster that regresses all 24 future values as a weighted sum of the past 168 values. We also implement DLinear, which decomposes the time series with a moving average kernel, applies a linear layer to each component, and then sums the two to get the final prediction.

**RNN (Gaussian):** This is an autoregressive encoder-decoder recurrent neural network inspired by Salinas et al. [38]. A multi-layer LSTM [18] first encodes the 168 past load values. The last encoder hidden state initializes the state of a multi-layer LSTM decoder. A linear layer maps each output of the decoder to Gaussian parameters.

## 5 Benchmark Results

Here, we analyze our baselines on the two benchmark tasks guided by the following questions:

1. Can models pretrained on Buildings-900K generalize to real buildings?
2. Does fine-tuning a pretrained model on limited data from a target building lead to improved performance?

Table 3: **Zero-shot STLF results.** Median accuracy (NRMSE) and ranked probability score (RPS) with **best** and second best highlighted. Lower is better. No fine-tuning is performed on any test building. Residential NRMSEs are naturally larger than commercial buildings, because the normalization factor—the building's average consumption per hour—is small). For example, Transformer-L (Gaussian) has an RMSE of 0.72 kWh and an NRMSE of 66.57% on the residential Sceaux dataset.

| | Buildings-900K-Test (*simulated*) | | | | BuildingsBench (*real*) | | | |
| | Commercial | | Residential | | Commercial | | Residential | |
| | NRMSE (%) | RPS | NRMSE (%) | RPS | NRMSE (%) | RPS | NRMSE (%) | RPS |
|---|---|---|---|---|---|---|---|---|
| Persistence Ensemble | 33.10 | 4.58 | 54.77 | 0.72 | 16.68 | 5.88 | **77.88** | **0.063** |
| Previous Day Persistence | 34.91 | - | 59.38 | - | 16.96 | - | 98.41 | - |
| Previous Week Persistence | 32.05 | - | 74.08 | - | 19.39 | - | 99.77 | - |
| Transformer-L (Tokens) | **12.91** | 2.45 | 44.29 | 0.875 | 14.46 | 5.62 | 95.34 | 0.152 |
| Transformer-L (Gaussian) | 15.10 | **2.07** | **43.52** | **0.578** | **13.31** | **5.23** | 79.34 | 0.072 |

Table 4: **Transfer learning results**. We show median accuracy (NRMSE) and ranked probability score (RPS) with **best** and second best highlighted. Transformer-S and Transformer-M results are in Fig. 4. Improvement due to fine-tuning.

| | Commercial buildings | | Residential buildings | |
| | NRMSE (%) | RPS | NRMSE (%) | RPS |
|---|---|---|---|---|
| **Not pretrained + Not fine-tuned** | | | | |
| Persistence Ensemble | 16.80 | 5.97 | 78.54 | **0.057** |
| Previous Day Persistence | 16.54 | - | 98.35 | - |
| Previous Week Persistence | 18.93 | - | 100.20 | - |
| **Not pretrained + Fine-tuned** | | | | |
| Linear regression | 25.18 | - | 89.98 | - |
| DLinear | 23.41 | - | 87.89 | - |
| RNN (Gaussian) | 41.79 | 15.28 | 96.75 | 0.078 |
| LightGBM | 16.02 | - | 80.07 | - |
| Transformer-L (Tokens) | 50.12 | 26.89 | 105.65 | 16.36 |
| Transformer-L (Gaussian) | 37.21 | 15.94 | 92.99 | 0.081 |
| **Pretrained + Not fine-tuned** | | | | |
| Transformer-L (Tokens) | 14.20 | 5.11 | 94.11 | 0.140 |
| Transformer-L (Gaussian) | 13.03 | 4.47 | 79.43 | 0.062 |
| **Pretrained + Fine-tuned** | | | | |
| Transformer-L (Tokens) | 14.07 (-0.13) | 4.99 (-0.12) | 94.53 (+0.42) | 0.137 (-0.003) |
| Transformer-L (Gaussian) | **12.96** (-0.07) | **4.37** (-0.10) | **77.20** (-2.23) | **0.057** (-0.015) |

3. How does the number of pretraining buildings affect zero-shot generalization?

4. How does the size of the pretrained model affect generalization?

## 5.1 Zero-Shot STLF

For zero-shot STLF, models are evaluated on day-ahead forecasting on all unseen simulated and real buildings *without fine-tuning*. Results for Transformer-L and other baselines are shown in Table 3 aggregated by the median over all buildings. Due to space constraints, Transformer-S and M results are shown in Figs. 3c-3d and App. Figs. 7-8. Performance profiles over all buildings (for comparing models by examining the tails of the forecast error distribution) and per-dataset metrics with 95% stratified bootstrap CIs are reported in App. H and I.

On unseen simulated buildings, the pretrained transformers outperform the persistence baselines in accuracy and uncertainty quantification. The average zero-shot STLF accuracy slightly improves **from simulated to real** commercial buildings (-0.1% NRMSE) and drops for residential buildings (+43% NRMSE). RPS is sensitive to the load magnitude, which makes it difficult to compute a sim-to-real gap for uncertainty quantification. See Fig. 2 for qualitative visualizations of the forecast

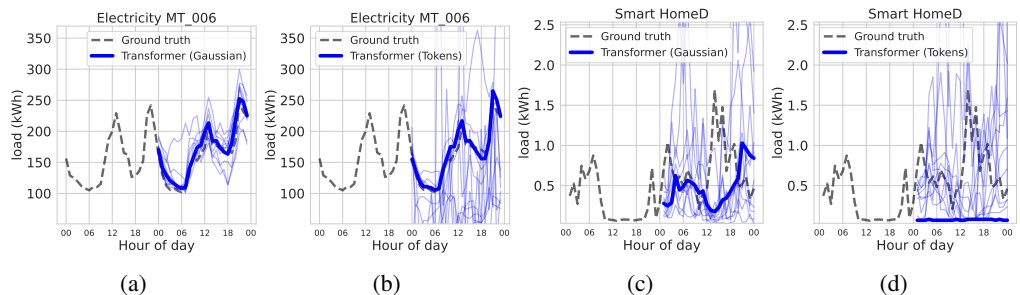

Figure 2: **Forecast uncertainty**. Ground truth time series are truncated to previous 24 hours for visibility. Light blue lines are 10 samples from the predicted distribution. a-b) Successful commercial building forecasts. c-d) Failed residential building forecasts.

uncertainty. The best pretrained model Transformer (Gaussian) outperforms the best persistence method Persistence Ensemble on real commercial buildings. However, Persistence Ensemble has better accuracy than the pretrained models on real residential buildings. These results suggest synthetic pretraining is viable for commercial zero-shot STLF—see Sec. 6.2 for more discussion on residential buildings.

## 5.2 Transfer Learning

This task evaluates fine-tuning on a single target building for which 6 months of data has been collected. Our fine-tuning protocol is to train for a max of 25 epochs using the first 5 months and to use the last month for early stopping with a patience of 2 epochs. To ease the computational burden of fine-tuning a model on each building in the benchmark, we randomly sample 100 residential and 100 commercial buildings and fine-tune separately on each. We fine-tune the pretrained transformers across all sizes and train randomly initialized transformers. Table 4 displays results aggregated by the median over all buildings; see Fig. 4 for the Transformer-S and Transformer-M results. We also report performance for the pretrained models without fine-tuning.

Our results indicate that fine-tuning the pretrained transformers on limited data is likely to improve the performance on the target building. For statistical robustness, we compute average probability of improving NRMSE due to fine-tuning: $P(X < Y) := \frac{100}{N} \sum_{i=1}^{N} \mathbf{1}_{[X_i < Y_i]}$ where $X_i$ is the pretrained + fine-tuned NRMSE for building $i$ and $Y_i$ is the pretrained NRMSE. The Transformer-M models have the highest $P(X < Y)$, with the Gaussian model achieving 98% and 73% respectively for commercial and residential buildings and the Tokens model scoring 70% and 62.5% (Fig. 4). For Transformer-L (Gaussian), this drops to 71% and 68% and to 61.5% and 39% for Transformer-L (Tokens). We found that fine-tuning *all* layers of both pretrained models was necessary—only fine-tuning the model's last layer led to slightly decreased performance (App. K). Encouragingly, the fine-tuned model (Transformer-L (Gaussian)) beats both LightGBM, a strong baseline in the low-data regime, and the Persistence Ensemble on both commercial and residential buildings.

## 5.3 Empirical Scaling Laws

**Pretraining buildings:** To characterize how size and diversity of the dataset impacts generalization, we trained the Transformer-M (17M) models on 1K, 10K, and 100K simulated buildings. We use early stopping to prevent the models trained on smaller datasets from overfitting. The Transformer-M model accuracy (Fig. 3a) and RPS (Fig. 3b) on commercial buildings roughly follows a power-law scaling with diminishing returns. Performance remains roughly constant across dataset scales for residential buildings (see Fig. 7 in App. J), which we attribute to the stronger sim-to-real distribution shift compared to commercial buildings. This suggests naively increasing the dataset size may not result in large improvements. Nevertheless, the best performance is achieved when using the full pretraining dataset of 900K buildings.

**Model size:** Here we compare the transformers of sizes S, M, and L. The NRMSE and RPS for real commercial buildings improves from S to M, but performance plateaus or decreases from the M to L models (Fig. 3c and Fig. 3d). We suspect this is due to distribution shifts and auto-correlation in the

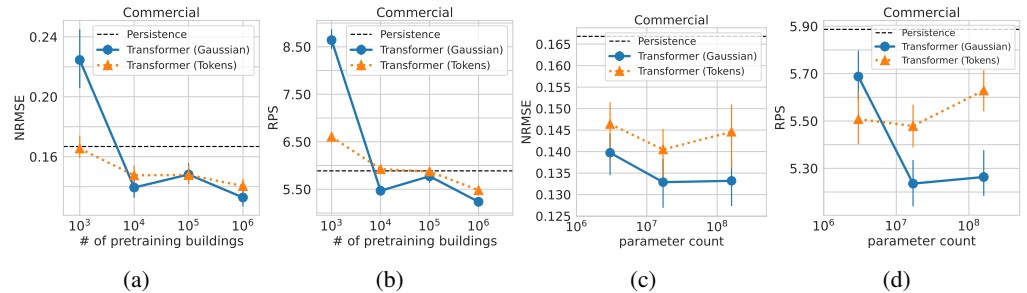

Figure 3: **Empirical scaling laws for zero-shot generalization on commercial buildings**. Intervals are 95% stratified bootstrap CIs for the median across all buildings. a-b) Dataset scale vs. zero-shot performance. The trends appear to be power-laws with diminishing returns. c-d) Model size vs. zero-shot performance. Residential results are in App. J.

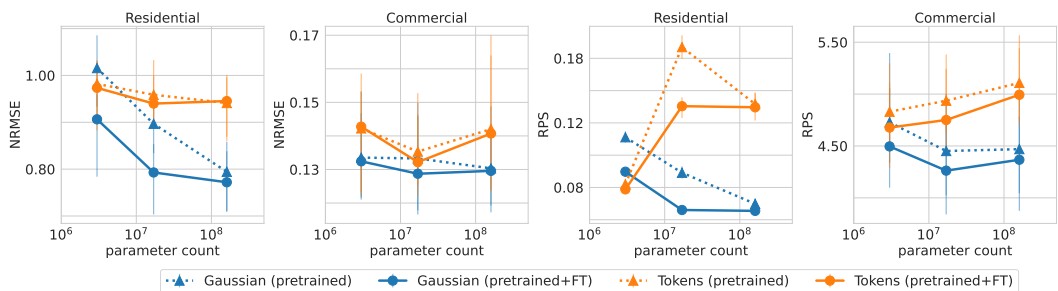

Figure 4: **Model size vs. transfer learning**. Pretrained vs. pretrained + fine-tuned (FT) performance for S, M, and L transformers. Intervals are 95% stratified bootstrap CIs of the median. The Transformer-M models show the most improvement after fine-tuning. The fine-tuned Transformer-M performance is comparable to the largest models. Improvement due to fine-tuning is less pronounced for the Transformer-L models, suggesting their zero-shot performance is saturated on this task.

time series that causes the largest models to overfit, despite using sliding windows to extract training samples and aggressive early stopping. The good RPS performance of the smallest Transformer (Tokens) model is likely due to training on quantized load values. Quantization helps generative models efficiently learn important structure in the data and ignore negligible information [37] (our tokenizer achieves a $\sim$63% dataset compression rate). Although, this does come at a cost of lower accuracy (Fig. 3c). While we show power-law scaling for residential building forecast accuracy (Fig. 7), the model performance may be saturated; see App. J for an in-depth discussion. We also show the relationship between model size and transfer learning performance in Fig. 4. The Transformer-M models demonstrate the largest performance gains due to fine-tuning, which confirms the high average probability of improvement scores (Sec. 5.2). The smaller impact of fine-tuning on Transformer-L performance suggests its zero-shot performance may be nearly saturated.

## 6 Discussion

### 6.1 Findings

**Pretraining on Buildings-900K leads to good zero-shot STLF performance on real commercial buildings**. More investigation is needed to achieve similar results for residential buildings (Sec. 6.2).

**Buildings-900K pretraining + fine-tuning on real buildings improves STLF for both commercial and residential buildings**. Our best pretrained + fine-tuned baseline outperforms LightGBM and persistence in the BuildingsBench transfer learning task. Fine-tuning all model layers appears necessary, possibly to mitigate negative transfer caused by the distribution shift between simulated pretraining and real test data.

**We observe an approximate power-law relationship with diminishing returns between dataset scale and zero-shot STLF for commercial buildings**. Performance plateaus as model size increases, possibly due to overfitting and distribution shifts.

**Pretraining on tokenized building loads instead of continuous loads achieves worse performance overall**. However, the tokenized model is more stable to train, as it naturally handles a wide range of load values.

**Pretraining with geospatial coordinates slightly improves generalization:** Due to space constraints, the results of an ablation where the model ignores the building's latitude and longitude are provided in App. K. Briefly, improvements in accuracy were modest (0.1 - 1% NRMSE).

## 6.2 Residential STLF Challenges

Residential loads are inherently more uncertain and variable than commercial loads because they are more sensitive to occupant behaviour and changes in weather (e.g., temperature and humidity) [39]. This is evidenced by the relatively large persistence NRMSEs (77.88%–99.77%) on BuildingsBench residential loads compared to commercial loads (16.68%–19.39%). However, rather than decreasing the value of BuildingsBench, our work enables exploring directions such as the inclusion of weather covariates, multi-variate formulations of STLF, and benchmarking of advanced approaches [20].

## 6.3 Limitations

While we expect BuildingsBench to stimulate research on generalizable STLF, our framework has limitations. First, pretraining on simulated data is fundamentally limited if deploying on real buildings is the goal. Mixing synthetic and real data during pretraining is an interesting direction to explore. Second, the pretraining and evaluation data is mainly representative of building energy consumption in the northwestern hemisphere. Expanding our framework to include data from other global regions is needed to comprehensively evaluate generalization. Third, the stochastic occupancy model used to simulate residential consumption behavior for the pretraining data is more predictable and less chaotic than real behavior, which increases the sim-to-real gap for residential buildings. Moreover, the occupancy model does not capture behavior patterns that may appear prominently in a specific geographic region (particularly outside of the U.S.), which may hurt the generalization capabilities when deploying the model in these locations. Finally, due to limited time, we only pretrained vanilla transformers on Buildings-900K. We encourage future work that compares these results with state-of-the-art transformers [31, 46]. We will maintain a leaderboard for BuildingsBench in our code repository, which will be updated with new model results.

# 7 Conclusions and Future Work

In this work, we introduce BuildingsBench, which consists of a large-scale simulated dataset and a collection of real building datasets for benchmarking zero-shot STLF and transfer learning. Upon comparing the performance of pretrained transformers against persistence and traditional machine learning-based forecasting baselines, we observe promising results for commercial buildings and identify areas of improvement for residential buildings.

We plan to maintain and extend BuildingsBench in the following ways. As relevant EULP data becomes newly available, we will release updated versions of Buildings-900K. To ground improvements to benchmark tasks in a concrete downstream application, we will look into adding a reinforcement learning task to the benchmark for building control using STLF. Other promising directions for future work include exploring joint forecasting of weather and load and the impact of building metadata on performance. To facilitate this work, we plan to update the datasets with this auxiliary information. Overall, we hope that BuildingsBench will facilitate research for communities applying machine learning to the built environment, as well as those conducting foundational studies on large-scale pretraining and fine-tuning for time series.

## Acknowledgments and Disclosure of Funding

This work was authored by the National Renewable Energy Laboratory (NREL), operated by Alliance for Sustainable Energy, LLC, for the U.S. Department of Energy (DOE) under Contract No. DE-AC36-08GO28308. This work was supported by the Laboratory Directed Research and Development (LDRD) Program at NREL. The views expressed in the article do not necessarily represent the views of the DOE or the U.S. Government. The U.S. Government retains and the publisher, by accepting the article for publication, acknowledges that the U.S. Government retains a nonexclusive, paid-up, irrevocable, worldwide license to publish or reproduce the published form of this work, or allow others to do so, for U.S. Government purposes. The research was performed using computational resources sponsored by the Department of Energy's Office of Energy Efficiency and Renewable Energy and located at the National Renewable Energy Laboratory.

The authors would like to thank Cong Feng and Alex Rybchuk for helping revise a draft version of this manuscript, as well as Dave Biagioni for providing valuable insights.

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

## Appendix

## Table of Contents

## A    Hosting, Licensing, and Maintenance

Our datasets and code are available via the following links:

- Github: `https://github.com/NREL/BuildingsBench`
- Documentation: `https://nrel.github.io/BuildingsBench`
- Datasets and Website: `https://data.openei.org/submissions/5859`
- Tutorials: `https://nrel.github.io/BuildingsBench/tutorials/`
- Dataset DOI: `https://doi.org/10.25984/1986147`

As described in Sec. 3 and Sec. 4, Buildings-900K and the BuildingsBench benchmark datasets are available for download under a CC-4.0 license and our code is available under a BSD 3-Clause license. We ensure the long-term availability and maintenance of the data by hosting it on the Open Energy Data Initiative (OEDI) platform (`https://data.openei.org/about`). The OEDI is a centralized repository for storing energy-related datasets derived from U.S. Department of Energy projects.

## B    Datasheet for Buildings-900K

Questions from Datasheet for Datasets (v8) [12].

### B.1    Motivation

**Q: For what purpose was the dataset created?**

This dataset was created to research large-scale pretraining of models for short-term load forecasting (STLF). It specifically addresses a lack of appropriately sized and diverse datasets for pretraining STLF models. Buildings-900K, which consists of simulated building energy consumption time series, is derived from the National Renewable Energy Lab (NREL) End-Use Load Profiles (EULP) database. We emphasize that the EULP was *not* originally developed for studying STLF. Rather, it was developed as a general resource to "...help electric utilities, grid operators, manufacturers, government entities, and research organizations make critical decisions about prioritizing research and development, utility resource and distribution system planning, and state and local energy planning and regulation." [45].

**Q: Who created the dataset (e.g., which team, research group) and on behalf of which entity (e.g., company, institution, organization)?**

Researchers at NREL created Buildings-900K. Consent from the NREL developers of the EULP project was obtained in writing to extract, process, and redistribute a subset of the EULP. The EULP is a multiyear multi-institutional collaboration lead by NREL and is publicly available for download with a permissive CC-4.0 license.

**Q: Who funded the creation of the dataset?**

This dataset was developed with funding provided by the Laboratory Directed Research and Development program at NREL.

### B.2    Composition

**Q: What do the instances that comprise the dataset represent (e.g., documents, photos, people, countries)?**

Annual hourly energy consumption from EnergyPlus [8] simulations. EnergyPlus is a high fidelity building energy consumption simulator which has been continuously maintained and updated for over 20 years by the U.S. Department of Energy.

**Q: How many instances are there in total (of each type, if appropriate)?**

There are ∼1.8 million annual time series instances (900K buildings × 2 years).

**Q: Does the dataset contain all possible instances or is it a sample (not necessarily random) of instances from a larger set?**

Buildings-900K consists of a subset of the EULP database. Specifically, we extract only the `out.site_energy.total.energy_consumption` time series from each `upgrade=0` building in the 2021 version for weather-years AMY2018 and TMY3. The EULP contains additional end-use time series for each building, the EnergyPlus XML model files, the EnergyPlus weather files, and a variety of additional simulations under various electrification scenarios. We did not include these additional data in the initial version of Buildings-900K, although we may consider adding them in the future.

**Q: What data does each instance consist of?**

Each instance is stored in a Parquet file and has timestamp information and load values (in kWh). We also have spatial metadata that maps each building to a U.S. county.

**Q: Is any information missing from individual instances?**

No.

**Q: Are relationships between individual instances made explicit (e.g., users' movie ratings, social network links)?**

Each building has a unique identifier and is assigned to a county. That county (or Public Use Microdata Area—PUMA) has its own unique identifier. Each PUMA contains many buildings that are geographically close to each other.

**Q: Are there recommended data splits (e.g., training, development/validation, testing)?**

- **For test:** We withhold all buildings in 4 counties from both AMY2018 and TMY3 weather-years.

- **For validation:** We withhold dates 2018-12-17 to 2018-12-31 from all buildings in the training set to use as held-out validation data (e.g., for early stopping).

- **For pretraining:** All remaining data is used for pretraining.

**Q: Are there any errors, sources of noise, or redundancies in the dataset?**

No.

**Q: Is the dataset self-contained, or does it link to or otherwise rely on external resources (e.g., websites, tweets, other datasets)?**

It is self-contained.

**Q: Does the dataset contain data that might be considered confidential (e.g., data that is protected by legal privilege or by doctor-patient confidentiality, data that includes the content of individuals' non-public communications)?**

No. All simulated building models are created by sampling from conditional distributions over attributes calibrated to real data.

**Q: Does the dataset contain data that, if viewed directly, might be offensive, insulting, threatening, or might otherwise cause anxiety?**

No.

**Q: Does the dataset relate to people?**

No.

### B.3 Collection Process

**Q: How was the data associated with each instance acquired?**

Buildings-900K time series instances are the outputs of extensively calibrated and validated EnergyPlus simulations. These simulations were created and run by the NREL EULP team on high performance computing resources. For completeness, we summarize here the steps taken by the EULP team to create the synthetic building models for these simulations, which are also described in the EULP documentation in more detail [45]. The synthetic buildings used to form large-scale U.S. residential and commercial building stock models were obtained from the ResStock[1] and ComStock[2] analysis tools. An extensive calibration and validation effort based on utility meter data from millions of customers, end-use submetering data, and additional private data sources helped to ensure the accuracy of the synthetic ResStock and ComStock models used in the EULP. A new stochastic occupant behavior model for the residential building stock was developed to help calibrate the simulations. Uncertainty quantification based on a learned surrogate [51] was also used for model calibration, in particular, to characterize sensitivity of simulation outputs (e.g., quantities of interest, or QOIs) with respect to different model parameters.

**Q: What mechanisms or procedures were used to collect the data (e.g., hardware apparatus or sensor, manual human curation, software program, software API)?**

See above.

**Q: If the dataset is a sample from a larger set, what was the sampling strategy (e.g., deterministic, probabilistic with specific sampling probabilities)?**

Buildings-900K consists of a hand-selected subset of the EULP database.

**Q: Who was involved in the data collection process (e.g., students, crowdworkers, contractors) and how were they compensated (e.g., how much were crowdworkers paid)?**

N/A

**Q: Over what timeframe was the data collected?**

A portion of Buildings-900K consists of building load time series simulated with weather from 2018.

**Q: Were any ethical review processes conducted (e.g., by an institutional review board)?**

No.

### B.4 Preprocessing/cleaning/labeling

**Q: Was any preprocessing/cleaning/labeling of the data done (e.g., discretization or bucketing, tokenization, part-of-speech tagging, SIFT feature extraction, removal of instances, processing of missing values)?**

The steps we followed to create Buildings-900K are as follows:

---

[1] https://resstock.nrel.gov
[2] https://comstock.nrel.gov

1. Select the `out.site_energy.total.energy_consumption` time series from each AMY2018 and TMY3 building.

2. Aggregate the 15-minute resolution energy consumption to hourly by *summing* the values at intervals of four consecutive timestamps XX:15, XX:30, XX:45, and XX+1:00 (e.g., 12:15, 12:30, 12:45, 13:00). Note that we use a sum aggregation here. When aggregating meter readings (in kWh) for real buildings (e.g., the BuildingsBench evaluation data), we take the *average* of the sub-hourly values.

3. Join the time series for all buildings in the same PUMA into a single Parquet table.

4. Save each PUMA-level Parquet table (compressed with Snappy).

**Q: Was the "raw" data saved in addition to the preprocessed/cleaned/labeled data (e.g., to support unanticipated future uses)?**

The raw data (the EULP) is publicly accessible for download `https://data.openei.org/s3_viewer?bucket=oedi-data-lake&prefix=nrel-pds-building-stock`.

**Q: Is the software used to preprocess/clean/label the instances available?**

Yes, at `https://github.com/NREL/BuildingsBench`.

### B.5 Uses

**Q: Has the dataset been used for any tasks already?**

In this work, we evaluate pretrained models on zero-shot STLF and transfer learning for STLF (fine-tuning pretrained models on limited data from a target building).

**Q: Is there a repository that links to any or all papers or systems that use the dataset?**

We will add links to papers or systems that use the dataset to `https://data.openei.org/submissions/5859` and `https://nrel.github.io/BuildingsBench`.

**Q: What (other) tasks could the dataset be used for?**

Pretrained models can sometimes serve as general-purpose feature extractors. These features can be used for a wide range of tasks, including:

- Predicting energy consumption of a building based on its characteristics.
- Classifying user behavior (e.g., whether a building is currently occupied) based on their energy consumption.
- Classifying building type based on energy consumption.
- Load disaggregation, that is, predicting the energy consumption of individual appliances based on the total energy consumption of a building.

Among others, potentially. We did not explore these tasks in this paper.

**Q: Is there anything about the composition of the dataset or the way it was collected and preprocessed/cleaned/labeled that might impact future uses?**

Buildings-900K is (in its current state) representative of buildings in the United States. While we find evidence that models pretrained on Buildings-900K generalize to buildings in other countries in the northwestern hemispheres, such as the UK and Canada, it may not be representative of building energy consumption patterns in other regions of the world.

**Q: Are there tasks for which the dataset should not be used?**

None that the authors are currently aware of.

### B.6 Distribution

**Q: Will the dataset be distributed to third parties outside of the entity (e.g., company, institution, organization) on behalf of which the dataset was created?**

Yes.

**Q: How will the dataset will be distributed (e.g., tarball on website, API, GitHub)?**

It will be publicly available for download on the Open Energy Data Initiative (OEDI) website at `https://data.openei.org/submissions/5859`.

**Q: ill the dataset be distributed under a copyright or other intellectual property (IP) license, and/or under applicable terms of use (ToU)?**

CC-4.0.

**Q: Have any third parties imposed IP-based or other restrictions on the data associated with the instances?**

No.

**Q: Do any export controls or other regulatory restrictions apply to the dataset or to individual instances?**

No.

### B.7 Maintenance

**Q: Who is supporting/hosting/maintaining the dataset?**

Code is hosted and publicly available on Github. NREL will maintain the code and datasets. The data is hosted on the Open Energy Data Initiative site, which is a centralized repository for energy-related datasets derived from U.S. Department of Energy projects.

**Q: How can the owner/curator/manager of the dataset be contacted (e.g., email address)?**

The lead maintainer at NREL is Patrick Emami (`Patrick.Emami@nrel.gov`).

**Q: Will the dataset be updated (e.g., to correct labeling errors, add new instances, delete instances)?**

Yes. A version naming system will be used to indicate updated versions. We are also maintaining a GitHub repository for the associated benchmark (BuildingsBench) where notifications of updates will be posted.

**Q: Will older versions of the dataset continue to be supported/hosted/maintained?**

Yes, older versions will remain available on the OEDI website.

**Q: If others want to extend/augment/build on/contribute to the dataset, is there a mechanism for them to do so?**

Not officially, but our benchmark code is open source and pull requests are welcome.

## C   BuildingsBench Datasets

### C.1   ElectricityLoadDiagrams20112014

This dataset is available from the UCI Machine Learning Repository[3] under a CC-4.0 license. It contains 15-minute consumption time series (in kW) for 370 clients in Portugal from 2011 to 2014. The magnitudes of the loads are much larger than residential homes, thus, we consider these time series as commercial buildings. After filtering for missing data, we kept 359 buildings. We included this dataset in BuildingsBench because of its permissive license, popular use in machine-learning-based time series forecasting studies [49], and size.

### C.2   The Building Data Genome Project 2

This dataset is available[4] under an MIT License. It consists of measurements from 3,053 meters from 1,636 commercial buildings over two years (2016 and 2017). One or more meters per building measured the total electrical, heating and cooling water, steam, solar energy, water, and irrigation

---

[3]`https://archive.ics.uci.edu/dataset/321/electricityloaddiagrams20112014`
[4]`https://github.com/buds-lab/building-data-genome-project-2/`

usage. This dataset was curated by the *Building Data Genome Project*, a consortium of academics and practitioners. We use the whole building electricity meter measurements from Bear, Fox, Panther and Rat sites, totalling 611 buildings (from the CSV file `electricity_cleaned.csv`). This dataset was included in BuildingsBench because of its permissive license, size, and diversity.

### C.3 Low Carbon London

This dataset is available[5] under a CC-4.0 license. It consists of consumption measurements from 5,567 London households that participated in a project led by UK Power Networks between November 2011 and February 2014. The dataset consists of half-hourly consumption (in kWh), a unique household identifier, and timestamps. Due to the large number of households included in the dataset, we randomly subsampled 713 of them (so as to keep the overall number of residential and commercial buildings in the benchmark approximately similar). We included this dataset because of its permissive license and size.

### C.4 SMART

This dataset is available online unaccompanied by a specific license[6]. It contains meter measurements of aggregate electricity data for 7 residential homes (which are called Home A,B,C,D,F,G,H) collected by researchers at UMass Amherst. After filtering out 2 homes due to missing data, we were left with Home B (years 2014–2016), Home C (2014–2016), Home D (2016), Home F (2014–2016), and Home G (2016). We included this dataset in BuildingsBench due to its accessibility and because the homes are located in the U.S. (Western Massachusetts), are are the simulated homes in Buildings-900K.

### C.5 IDEAL

This dataset is available[7] under a CC-4.0 license. It has meter readings for electric and gas usage, as well as room temperature, humidity, and boiler readings from 255 UK homes. We use the total home electricity measurements. The measurements were collected by researchers at the University of Edinburgh. After our filtering process, we kept 219 homes. This dataset was included in BuildingsBench due to its permissive license and size.

### C.6 Individual household electric power consumption (Sceaux)

This dataset is available via Kaggle[8] under a DbCL-1.0 license. It contains consumption measurements from a home in Sceaux, France between December 2006 and November 2010. It provides 7 types of meter readings at one minute resolution with timestamps. We use the global active power readings. This dataset was included in BuildingsBench due to its permissive license and due to its popularity both in the literature [28] and on Kaggle.

### C.7 Borealis

This dataset is available[9] under a CC0-1.0 license. It is 6-second resolution measurements of total real power consumption for 25 households in Waterloo, ON for at least three months between 2011 and 2012. After filtering out homes with excessive missing data, we kept 15 of them. The data was collected by researchers at the University of Waterloo. We included this dataset in BuildingsBench due to its permissive license and size.

### C.8 Outlier Removal

Here we provide additional details on outlier removal for noisy meter data. The algorithm we use compares each load value to all others in a sliding window of 24 hours and computes the absolute difference with respect to the 1-nearest neighbor (1-NN). To select an outlier threshold, we compute

---

[5]`https://data.london.gov.uk/dataset/smartmeter-energy-use-data-in-london-households`
[6]`https://traces.cs.umass.edu/index.php/smart/smart`
[7]`https://datashare.ed.ac.uk/handle/10283/3647`
[8]`https://www.kaggle.com/datasets/uciml/electric-power-consumption-data-set`
[9]`https://borealisdata.ca/dataset.xhtml?persistentId=doi:10.5683/SP2/R4SVBF`

the average daily peak load and average daily base load, and compute the difference. If the 1-NN distance is larger than this difference, we classify the spike as an outlier and replace the spike with the median of the sliding window.

# D  Evaluation Metrics

## D.1  Additional Accuracy Metrics: NMAE, NMBE

**NMAE:** Similar to the NRMSE, the normalized mean absolute error (NMAE) measures the ability of a model to predict the correct load shape. However, it is less sensitive to large errors than the NRMSE. For a building with $M$ days of load time series,

$$NMAE := 100 \times \frac{1}{\bar{y}} \left[ \frac{1}{24M} \sum_{j=1,i=1}^{M,24} |y_{i,j} - \hat{y}_{i,j}| \right]. \tag{5}$$

**NMBE:** The normalized mean bias error (NMBE) is informative about a model's tendency to over or under-estimate building loads, which is practically useful to measure. However, a model that tends to over-estimate or under-estimate the load by an equal amount can achieve an NMBE close to zero (i.e., positive and negative errors cancel). Therefore, it is not a useful indicator of *accuracy*. For a building with $M$ days of load time series,

$$NMBE := 100 \times \frac{1}{\bar{y}} \left[ \frac{1}{24M} \sum_{j=1,i=1}^{M,24} (y_{i,j} - \hat{y}_{i,j}) \right]. \tag{6}$$

We visualize performance profiles and dataset-specific results for all metrics in App. H and App. I, respectively.

## D.2  Categorical Ranked Probability Score

Here, we define the ranked probability score (RPS) for categorical distributions. This is used to compute the RPS for our time series transformer variant that predicts a categorical distribution over a vocabulary of $|\mathcal{V}|$ discretized load tokens. Each load token is the centroid of a cluster, where clusters are estimated via KMeans. See App. E for more details about the tokenizer.

In what follows, we assume that the vocabulary of load tokens is sorted in increasing order by value, that is, $0 \leq v_0 < v_1 < \cdots < v_{|\mathcal{V}|-1}, v_i \in \mathcal{V}$.

Let $p(\hat{Y}_{i,j})$ be the predicted categorical distribution over $\mathcal{V}$ for hour $i$ on day $j$. We assume this is a normalized distribution, which is accomplished in practice by computing the softmax of the model's predicted logits. The corresponding CDF for $p(\hat{Y}_{i,j})$ is $F(\hat{Y}_{i,j})$, which is given by the cumulative sum of the $|\mathcal{V}|$ sorted entries of $p(\hat{Y}_{i,j})$. Then, let $F(Y_{i,j})$ be the CDF of the ground truth discretized load. If the ground truth category is $k \in \{0, \ldots, |\mathcal{V}| - 1\}$, the CDF $F(Y_{i,j})$ is 0 up to index $k - 1$ and 1 thereafter (resembling a step function that goes from 0 to 1 at ground truth load index $k$). Finally, define $\bar{v}[k]$ as the difference between the maximum load value and the minimum load value assigned to load token $v[k]$ by KMeans (recall, load tokens are centroids of 1D KMeans clusters).

Then, the categorical RPS at hour $i$ for a building with $M$ days of load time series is

$$CatRPS_i := \frac{1}{M} \sum_{j=1}^{M} \sum_{k=0}^{|\mathcal{V}|-1} (F(\hat{Y}_{i,j})[k] - F(Y_{i,j})[k])^2 \bar{v}[k]. \tag{7}$$

## D.3  Gaussian Ranked Probability Score

The *continuous* ranked probability score for Gaussian distributions can be evaluated in closed form. In particular, let $\hat{\mu}_{i,j}$ and $\hat{\sigma}_{i,j}$ be a predicted mean and standard deviation, and $y_{i,j}$ be the ground truth value. Define $z_{i,j} = \frac{y_{i,j} - \hat{\mu}_{i,j}}{\hat{\sigma}_{i,j}}$. Then, the Gaussian (C)RPS at hour $i$ for a building with $M$ days

of load time series is

$$GaussCRPS_i := \frac{1}{M} \sum_{j=1}^{M} \hat{\sigma}_{i,j} \left[ z_{i,j} \left( 2 \underbrace{\left( \frac{1 + \text{erf}(z_{i,j})}{2\sqrt{2}} \right)}_{\text{Gaussian CDF}} - 1 \right) + 2 \underbrace{\left( \frac{\exp{\frac{-z_{i,j}^2}{2}}}{\sqrt{2\pi}} \right)}_{\text{Gaussian PDF}} - \frac{1}{\sqrt{\pi}} \right]. \quad (8)$$

**Gaussian approximation of the inverse Box-Cox transform:** Our time series transformer variant that predicts a Gaussian distribution for each hour $i$ in the day-ahead forecast uses a Box-Cox power transformation to normalize the load values. In App. K, we show that standard scaling was insufficient to remove training instabilities caused by the non-Gaussian load time series. Since the Gaussian distribution is learned in the *Box-Cox transformed space*, we cannot directly use the GaussCRPS metric to evaluate uncertainty quantification *in the original un-scaled space* (i.e., with units of kWh). Our current solution, introduced here, is to compute a Gaussian distribution in the original un-scaled space that closely approximates the distribution given by the inverse Box-Cox transformation (for a reasonable range of standard deviations). We leave exploring advanced non-approximate solutions for future work.

First, we define the Box-Cox transformation of a random variable $X$

$$Y = f_\lambda(X) := \begin{cases} \frac{X^\lambda - 1}{\lambda} & \text{if } \lambda \neq 0 \\ \ln(X) & \text{if } \lambda = 0, \end{cases} \quad (9)$$

and its inverse

$$X = f_\lambda^{-1}(Y) := \begin{cases} (Y\lambda + 1)^{\frac{1}{\lambda}} & \text{if } \lambda \neq 0 \\ \exp(Y) & \text{if } \lambda = 0. \end{cases} \quad (10)$$

We estimate a Gaussian distribution in the un-scaled space that *approximates* the (power-normal) distribution of the samples inverted with $f_\lambda^{-1}$. Recall that $\hat{\mu}_{i,j}$ and $\hat{\sigma}_{i,j}$ are the predicted mean and standard deviation in the Box-Cox transformed space. We take the inverse of the predicted mean to be $f_\lambda^{-1}(\hat{\mu}_{i,j})$ (Eq. 10). Then, the standard deviation is estimated with:

$$\hat{\sigma}_{i,j}^+ = f_\lambda^{-1}(\hat{\mu}_{i,j} + \hat{\sigma}_{i,j}) - f_\lambda^{-1}(\hat{\mu}_{i,j}) \quad (11)$$

$$\hat{\sigma}_{i,j}^- = f_\lambda^{-1}(\hat{\mu}_{i,j}) - f_\lambda^{-1}(\hat{\mu}_{i,j} - \hat{\sigma}_{i,j}) \quad (12)$$

$$\tilde{\sigma}_{i,j} \approx \frac{\hat{\sigma}_{i,j}^+ + \hat{\sigma}_{i,j}^-}{2}. \quad (13)$$

The Gaussian distribution in the un-scaled space is thus $\mathcal{N}(f_\lambda^{-1}(\hat{\mu}_{i,j}), \tilde{\sigma}_{i,j}^2)$. We find that this Gaussian is a good approximation for the distribution of the samples obtained by applying Eq. 10 to samples drawn from $\mathcal{N}(\hat{\mu}_{i,j}, \hat{\sigma}_{i,j}^2)$ for reasonably small values of $\tilde{\sigma}_{i,j}$. We compare these two distributions in Fig. 5. When the model is uncertain about the forecast and the standard deviation in the un-scaled space $\tilde{\sigma}_{i,j}$ is large, the true distribution in the un-scaled space is heavily skewed to the right, making a Gaussian a poor approximation (Fig. 5c).

## E  Time Series Transformer Details

Table 5: Transformer architecture hyperparameters.

|  | Predicted Distribution | # Layers | # Heads | Embedding Dim | MLP Dim | # Params |
|---|---|---|---|---|---|---|
| Transformer-S | Categorical | 2 | 4 | 256 | 512 | 3.3M |
| Transformer-M | Categorical | 3 | 8 | 512 | 1024 | 17.2M |
| Transformer-L | Categorical | 12 | 12 | 768 | 2048 | 160.7M |
| Transformer-S | Gaussian | 2 | 4 | 256 | 512 | 2.6M |
| Transformer-M | Gaussian | 3 | 8 | 512 | 1024 | 15.8M |
| Transformer-L | Gaussian | 12 | 12 | 768 | 2048 | 160.7M |

In this section, we provide additional details about the time series transformer architecture used in this work. We use the original encoder-decoder model from Vaswani et al. [43] as implemented in the

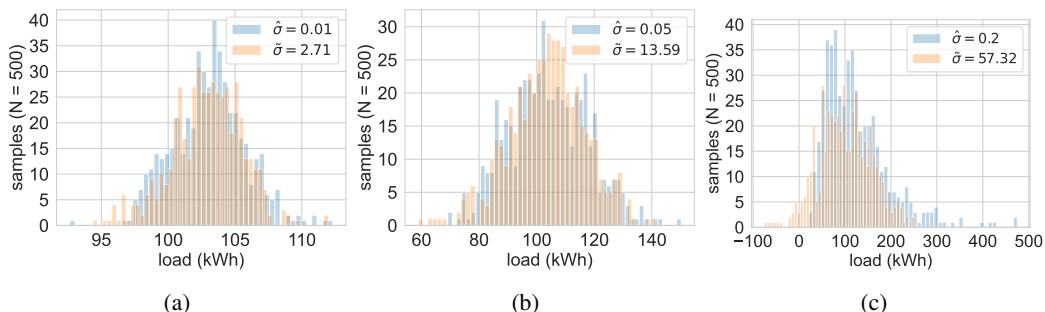

(a)  (b)  (c)

Figure 5: **Gaussian approximation of the inverse Box-Cox**. Visualization of the Gaussian distribution $\mathcal{N}(f^{-1}(\hat{\mu}_{i,j}), \tilde{\sigma}_{i,j})$ (orange) in the un-scaled space of loads (in kWh) for computing the RPS when using Box-Cox scaling. For reasonably small standard deviations $\tilde{\sigma}$ in the un-scaled space, this Gaussian is a reasonable approximation of the power-normal distribution given by the inverse Box-Cox (blue). c) When the model is highly uncertain so that $\tilde{\sigma}$ is large, the power-normal is extremely right-skewed. In this case, the Gaussian approximation is inaccurate.

PyTorch `nn.Transformer` module. This autoregressive model is trained to predict the load at each time step with teacher forcing. Following Wu et al. [47], we use masking in the decoder to prevent the attention from peeking at the target value at each time step. The first value in the input sequence passed to the decoder is the last value of the input sequence passed to the encoder. The Gaussian time series transformer uses a linear layer (with parameters shared across time steps) to transform the high-dimensional output of the decoder into a mean and standard deviation at each time step. The time series transformer trained on tokenized load values uses a linear layer to transform the decoder outputs into $|\mathcal{V}|$ logits for the cross-entropy loss, also shared across time steps. Our tokenizer is based on KMeans (a stochastic algorithm) followed by a token merging step. Depending on the random seed, multiple runs of the tokenizer will produce token vocabularies of different sizes. The tokenizer is described in detail in App. E.1.

The architecture hyperparameters for the L, M, and S models are shown in Table 5. The largest model has 12 layers, 12 heads, and an internal dimension of 768 selected to match Radford et al. [35], but uses a smaller feedforward dimension of 2048 to keep the number of parameters at approximately $10^8$. We downsize these architecture hyperparameters to reduce the number of parameters by roughly an order of magnitude for the S and M models. Similar to Radford et al. [35], all models use GELU [16] activations and $\mathcal{N}(0, 0.02)$ weight initialization.

**Model input encoding:** Let $E$ be the embedding dimension and $s$ be a scaling factor given by `E`/256 (e.g., $s = 3$ for Transformer-L). The model inputs are encoded, concatenated, and passed to the transformer encoder as follows:

- **day of year**, **day of week**, **hour of day**: These are encoded into a 2-dim space with $\sin(\pi x)$ and $\cos(\pi x)$ then projected with a linear layer to $32s$-dim.
- **latitude, longitude**: These scalars are each projected to $32s$-dim space with linear layers.
- **building type**: A linear embedding layer is used to learn a projection for each of residential and commercial building types to $32s$-dim space.
- **continuous loads**: When the load is a continuous scalar, it is projected to $64s$-dim space with a linear layer.
- **discrete load tokens**: When the load is a discrete token, a linear embedding layer is used to learn a projection for each of the $|\mathcal{V}|$ tokens to $64s$-dim space.

These various embeddings are concatenated into a feature vector of size $E$ at each time step.

### E.1 Load Tokenizer

We explore quantizing load values into a vocabulary of discrete tokens to closely mimic the application of transformers to natural language. The design of the tokenizer is detailed here. We use simple KMeans clustering with a basic merging strategy inspired by Byte Pair Encoding [11] which merges

**Algorithm 1** Load tokenizer

```
// Inputs:  loads ((n,1) array), K initial centroids, τ = 0.01 threshold (kWh)
kmeans = faiss.Kmeans(K)
kmeans.train(loads)
// Merge step.  Initialize state.
sorted_centroids = sort(kmeans.centroids)
current_centroid = sorted_centroids[0]
merged_centroids = []
temp_centroids = [current_centroid]
// Iterate over sorted centroids in increasing order
for i = 1 to K do
  // The next centroid is less than τ away, merge it
  if sorted_centroids[i] - current_centroid < τ then
    temp_centroids.append(sorted_centroids[i])
  else
    merged_centroids.append(mean(temp_centroids))
    temp_centroids = [sorted_centroids[i]]
    current_centroid = sorted_centroids[i]
// Add the last centroid to the new list of merged centroids
merged_centroids.append(mean(temp_centroids))
return kmeans, merged_centroids
```

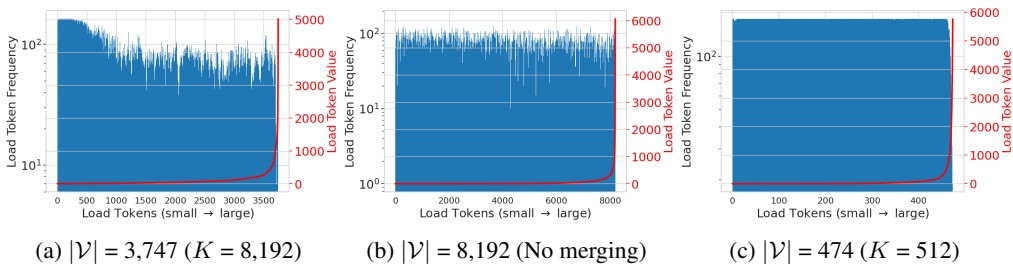

(a) $|\mathcal{V}| = 3{,}747$ ($K = 8{,}192$)  (b) $|\mathcal{V}| = 8{,}192$ (No merging)  (c) $|\mathcal{V}| = 474$ ($K = 512$)

Figure 6: **Load token frequency**. We visualize histograms of token frequencies from a subsample of Buildings-900K. a) After merging $K = 8{,}192$ into 3,747, there are some tokens that appear often near load values of zero while most other tokens have nearly equal usage. The mean absolute quantization error for commercial buildings is approximately 0.2 kWh whereas for residential buildings it is 0.003 kWh. b) Without merging, most tokens correspond to small load values and all appear to have roughly equal usage. c) When $|\mathcal{V}|$ is small (474), there are few tokens for large load values.

clusters that are overly close to each other. We leave exploring other simple and sophisticated tokenization strategies for future work.

The tokenizer uses KMeans clustering on a random subsample of load values from the Buildings-900K training set to fit $K$ clusters (the size of the subsample is chosen internally by `faiss-gpu`). The merge step is as follows. First, we sort the $K$ cluster centroids in increasing order and then iterate over them. For any cluster centroid $i$, if there are successive centroids that are less than $\tau = 0.01$ kWh greater than $i$, we replace all of these centroids with a new centroid equal to their average. We provide pseudocode in Alg. 1. This algorithm is stochastic, such that running the tokenizer multiple times will produce vocabularies with different (but similar) sizes $|\mathcal{V}|$. To tokenize a continuous load value, we first look up the index of the original centroid it belongs to, then use this index to look up the corresponding index in the list of merged centroids. See Fig. 6 for a visual comparison of three different vocabularies produced by the tokenizer, and App. K.1 for ablation study results on the choice of $K$ and the merging step.

# F   Training Details

In this section, we describe how training hyperparameters were selected for the transformer models. We use the loss on the Buildings-900K validation dataset for model selection. However, we observed

that a better Buildings-900K validation loss does not necessarily indicate better downstream performance on the real (out-of-distribution) evaluation data. We encourage exploring sophisticated model selection strategies that account for distribution shifts in future work.

**Pretraining hyperparameters:** The learning rate for all models uses a linear warmup of 10K steps followed by cosine decay to zero. Models are trained on 1 billion load hours, as we observed in early runs that the validation loss would no longer improve after this point. We perform a grid search over the max learning rate {6e-4, 6e-5, 6e-6} and the batch size {64, 128, 256} for the two Transformer-S models. For the Transformer-S (Gaussian) model, the highest learning rate and smallest batch size had the best validation loss. However, when applying these hyperparameters to the larger models, the high learning rate of 6e-4 caused training instability. Thus, we used the lower 6e-5 learning rate, which performed best with the smallest batch size of 64 (650K total gradient updates). Similarly for the Transformer-S (Tokens) model, the highest learning rate achieved the best validation loss, but for the larger models we used the lower 6e-5 learning rate with batch size of 64. We use the AdamW [26] optimizer with $\beta_1 = 0.9$, $\beta_2 = 0.98$, $\epsilon = 1e-9$, and weight decay of 0.01 following Radford et al. [35].

**Transfer learning hyperparameters:** We fine-tune all layers of the transformers (as opposed to only the last logits layer—see App. K.4 for an ablation study on this) with a lower learning rate of 1e-6. We allow a maximum of 25 fine-tuning epochs with an early stopping patience of 2. We tune the learning rate for the Linear, DLinear, and RNN baselines by sweeping over 5 values: {1e-2, 1e-3, 1e-4, 1e-5, 1e-6}. All non-pretrained models are trained for a maximum of 100 epochs.

## G   Compute Details

- **Buildings-900K processing:** Creating the Buildings-900K dataset by processing a subset of the EULP was accomplished with a 96-core AWS EC2 instance in 2-3 days.

- **Model pretraining:** The largest model variant, the Transformer-L, was trained on 1 billion hours from Buildings-900K with one NVIDIA 40GB A100 GPU in 24 hours. Smaller model variants train faster, in about 18 hours. We use PyTorch automatic mixed precision training with gradient scaling, which is crucial for achieving fast training times.

- **Zero-shot STLF:** The amount of time to run each benchmark task varies depending on model inference speed. This task uses all available years for every building in the benchmark. For the largest model variant, it takes about 2-3 hours on one NVIDIA 40GB A100. By contrast, the persistence baselines run in well under one hour.

- **Transfer learning:** Repeating the fine-tuning process for every building in the benchmark can be computationally demanding. For example, we sub-sample 200 total buildings (100 residential and 100 commercial) to use for this task, and this takes the Transformer-L models 1.5-3.5 hours to complete on one NVIDIA 40GB A100. We estimate it would take over 15 hours to include all buildings in this task. Models that are fine-tuned from frozen pretrained weights finish this task faster, as we stop the fine-tuning process with early stopping using a patience of 2 epochs.

## H   Performance Profiles

Performance profiles plot the fraction of all buildings with performance greater than a threshold $\tau$ for a range of threshold values. These can be used to understand the tails of the error distributions across all buildings in the benchmark. For example, these plots show the fraction of buildings with extremely high forecasting errors.

Here we show performance profiles for commercial (Fig. 12, Fig. 13) and residential (Fig. 14, Fig. 15) BuildingsBench buildings for the NRMSE, NMAE, NMBE, and RPS metrics. For reference, we also plot the performance profile of the Persistence Ensemble baseline. The methods that estimate the forecast with a Gaussian distribution (the Persistence Ensemble and Transformer (Gaussian)) achieve noticeably better NMBE performance than Transformer (Tokens).

## I   Benchmark Dataset Results

Per-dataset zero-shot accuracy (NRMSE, NMAE, NMBE) are visualized in Fig. 16.

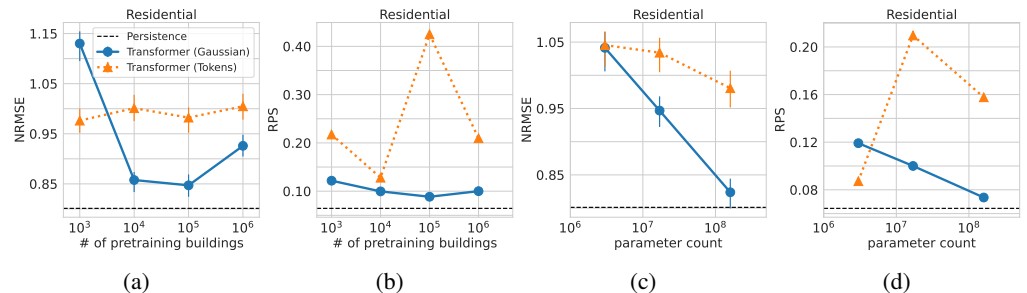

Figure 7: Empirical scaling laws for zero-shot generalization on residential buildings. Intervals are 95% stratified bootstrap CIs of the median. a-b) Dataset size vs. zero-shot performance. c-d) Model size vs. zero-shot performance.

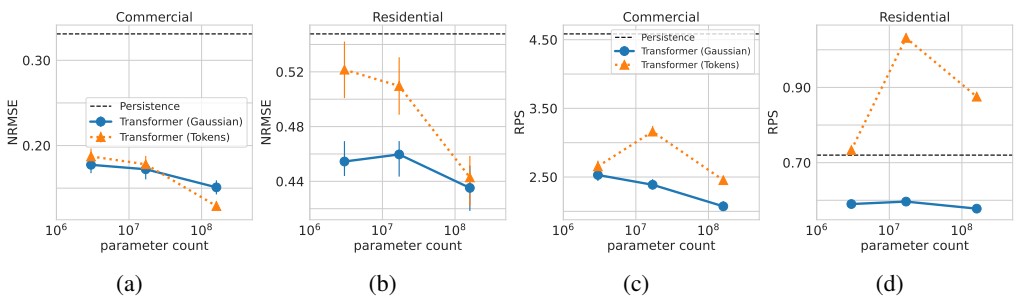

Figure 8: Model size vs. Buildings-900K-Test zero-shot STLF performance. Intervals are 95% stratified bootstrap CIs of the median. The trends in (a)-(c) suggest superior generalization to the test data is achievable by larger models, which might be unsurprising due to the *small* distribution shift between the simulated pretraining and simulated test data (relative to the *large* distribution shift between the simulated pretraining and real test data).

## J Additional Empirical Scaling Law Results

The residential building results for the experiments where the Transformer-M models are trained under increasing dataset scale (size and number of unique pretraining buildings) and where model scales are compared are shown in Fig. 7. In most cases, the accuracy (NRMSE) and uncertainty quantification (RPS) are independent of the dataset and model size. Exceptions are the Transformer-M (Gaussian) performance on the smallest pretraining dataset size and the effect of model size on the Transformer (Gaussian), which appears to show a power-law scaling. By contrast, Fig. 3 shows that the smallest Transformer-S (Gaussian) model achieves comparable performance to the larger variants on commercial buildings. A possible simple explanation is that the small model struggles to learn to jointly forecast for both residential *and* commercial buildings due to limited capacity. A key benefit of the larger model, then, is unlocking the ability to jointly forecast for both types of buildings. As discussed in Sec. 6.2, alternative modeling approaches that consider auxiliary inputs might be more successful for large-scale pretraining of residential building STLF. We believe this is an interesting future research direction. We also examine empirically how performance on the simulated Buildings-900K-Test varies with model scale (Fig. 8), and observe trends that clearly indicate superior generalization is achievable with larger models in this setting.

## K Ablation Studies

We explore the following aspects of the transformer models in this section:

- App. K.1) The size of the vocabulary and the impact of the merging step in the load tokenizer.
- App. K.2) Using standard scaling vs. Box-Cox scaling for training the Transformer (Gaussian) model.
- App. K.3) Impact of the lat, lon covariate on generalization.

Table 6: **Ablation study results on the zero-shot STLF task.** Median accuracy (NRMSE) and ranked probability score (RPS). Results are for Transformer-L (Tokens). We ablate the use of the lat,lon covariate and the size of the token vocabulary $|\mathcal{V}|$. The vocabulary with 8,192 tokens does not use merging (note the high RPS for residential buildings).

| | | Buildings-900K-Test (*simulated*) | | | | BuildingsBench (*real*) | | | |
| | | Commercial | | Residential | | Commercial | | Residential | |
| $|\mathcal{V}|$ | Lat,Lon | NRMSE (%) | RPS | NRMSE (%) | RPS | NRMSE (%) | RPS | NRMSE (%) | RPS |
|---|---|---|---|---|---|---|---|---|---|
| 3,747 | ✗ | 15.22 | 2.79 | 45.85 | 0.913 | 14.40 | 5.73 | 97.84 | 0.199 |
| 474 | ✓ | 12.70 | 1.62 | 42.94 | 0.544 | 14.71 | 5.16 | 94.75 | 0.113 |
| 8,192 | ✓ | 15.06 | 2.85 | 45.50 | 7.05 | 14.62 | 5.90 | 99.88 | 14.37 |
| 3,747 | ✓ | 12.91 | 2.45 | 44.29 | 0.875 | 14.46 | 5.62 | 95.34 | 0.152 |

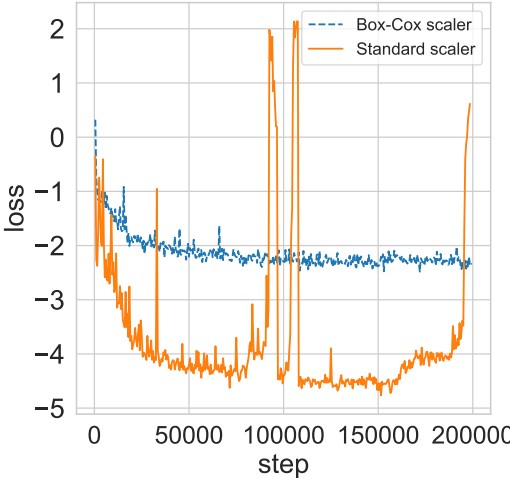

Figure 9: **Box-Cox scaling vs. standard scaling training stability**. Standard normalizing the load values did not prevent the Transformer (Gaussian) model from suffering from training instability. Box-Cox scaling was successful at achieving stable training.

- App. K.4) Fine-tuning all layers vs. only the last layer for transfer learning.

### K.1 Load Tokenizer

Our experiments are conducted with a merged vocabulary of size $|\mathcal{V}| = 3{,}747$ ($K = 8{,}192$). To investigate the impact of using a smaller vocabulary, we trained the Transformer-L model with a merged vocabulary of size $|\mathcal{V}| = 474$ ($K = 512$) (Fig. 6c). We also trained a model *without merging*, thus, $|\mathcal{V}| = 8{,}192$ ($K = 8{,}192$) (Fig. 6b). The results in Table 6 indicate that when the vocabulary is too small ($|\mathcal{V}| = 474$), the accuracy on real commercial buildings somewhat deteriorates. Without merging ($|\mathcal{V}| = 8{,}192$), the RPS on real residential buildings is significantly worse. Nearly all tokens correspond to small load values, which causes the model to allocate probability to a large number of incorrect tokens whose load values are very close to the target value.

### K.2 Standard Scaling Data Normalization

As can be seen in Fig. 9, global standard normalization was not sufficient to prevent training instability (large spikes in the loss) due to the highly non-Gaussian load values. By contrast, Box-Cox scaling was successful at stabilizing training.

### K.3 Latitude, Longitude Covariate

In our experiments, the transformer models have a latitude, longitude covariate that roughly localizes where a building is in the world. The result of training Transformer-L (Tokens) without this covariate

Table 7: **Ablation study results on transfer learning task**. Median accuracy (NRMSE) and ranked probability score (RPS). For statistical robustness, we compute average probability of improving NRMSE due to fine-tuning: $P(X < Y) := \frac{100}{N} \sum_{i=1}^{N} \mathbf{1}_{[X_i < Y_i]}$ where $X_i$ is the pretrained + fine-tuned NRMSE for building $i$ and $Y_i$ is the pretrained NRMSE. We compare fine-tuning only the last layer and fine-tuning all layers.

| | Commercial buildings | | | Residential buildings | | |
|---|---|---|---|---|---|---|
| | NRMSE (%) | RPS | $P(X < Y)$ | NRMSE (%) | RPS | $P(X < Y)$ |
| **Fine-tune last layer** | | | | | | |
| Transformer-L (Tokens) | 14.18 | 5.07 | 53.5 | 93.51 | 0.136 | 47 |
| Transformer-L (Gaussian) | 12.93 | 4.50 | 51 | 79.57 | 0.063 | 36 |
| **Fine-tune all layers** | | | | | | |
| Transformer-L (Tokens) | 14.07 | 4.99 | 61.5 | 94.53 | 0.137 | 39 |
| Transformer-L (Gaussian) | 12.96 | 4.37 | 71 | 77.20 | 0.057 | 68 |

are in Table 6. The buildings in the Buildings-900K-Test set are from held-out PUMAs (i.e., counties) with never-before-seen latitude, longitude coordinates. Without latitude and longitude covariates, we observe a negligible drop in zero-shot accuracy on real buildings, but a significant loss in accuracy on the *simulated* Buildings-900K-Test split. As the majority of real buildings are located in non-U.S. out-of-distribution locations, the latitude and longitude covariate appears to not be particularly helpful for generalizing to these buildings. This is despite mapping the non-U.S. latitude longitude coordinates to U.S. locations with similar climates, which we do to help avoid introducing errors due to extrapolation. We believe further study on the geospatial aspects of generalizable building STLF is a promising use-case of our benchmark.

### K.4 What Layers Get Fine-tuned?

For our main transfer learning benchmark results, we fine-tune all layers of the transformers. Here, we compare the performance of these models with fine-tuning only the last layer. Table 7 shows that when fine-tuning only the last linear layer (that output logits or Gaussian parameters), the performance suffers, which can be best seen by comparing the respective probabilities of improvement. Our intuition is that since the pretraining data is synthetic, fine-tuning on real data may require updating all layers. We believe a deeper investigation into transfer learning paradigms for time series is an important topic of study.

## L Author Responsibility Statement

The authors confirm that they bear all responsibility in case of any violation of rights during the collection of the data or other work, and will take appropriate action when needed, e.g. to remove data with such issues. The authors also confirm the licenses provided with the data and code associated with this work.

# M   Additional Plots

We provide additional qualitative results here (Fig. 10 and Fig. 11).

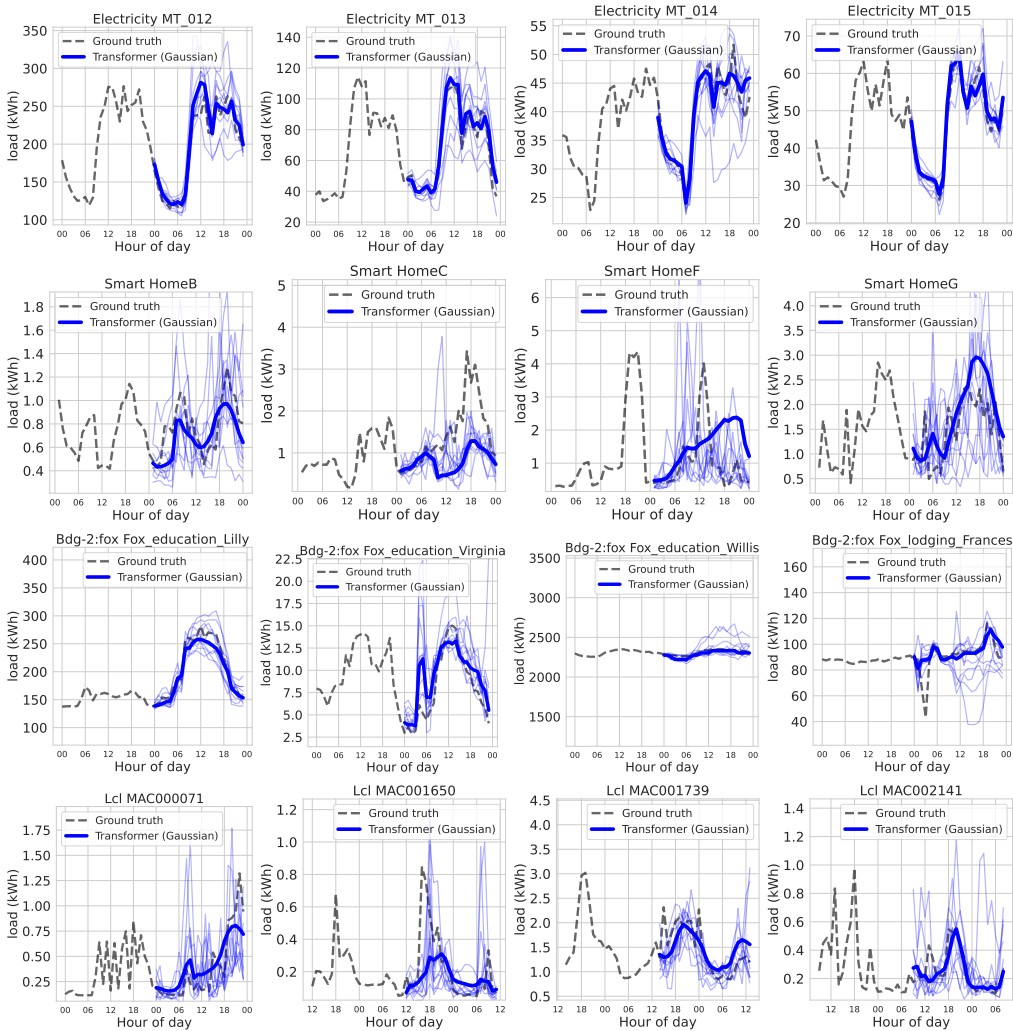

Figure 10: **Qualitative results for Transformer-L (Gaussian)**. Ground truth time series are truncated to previous 24 hours for visibility. Light blue lines are 10 samples from the predicted distribution.

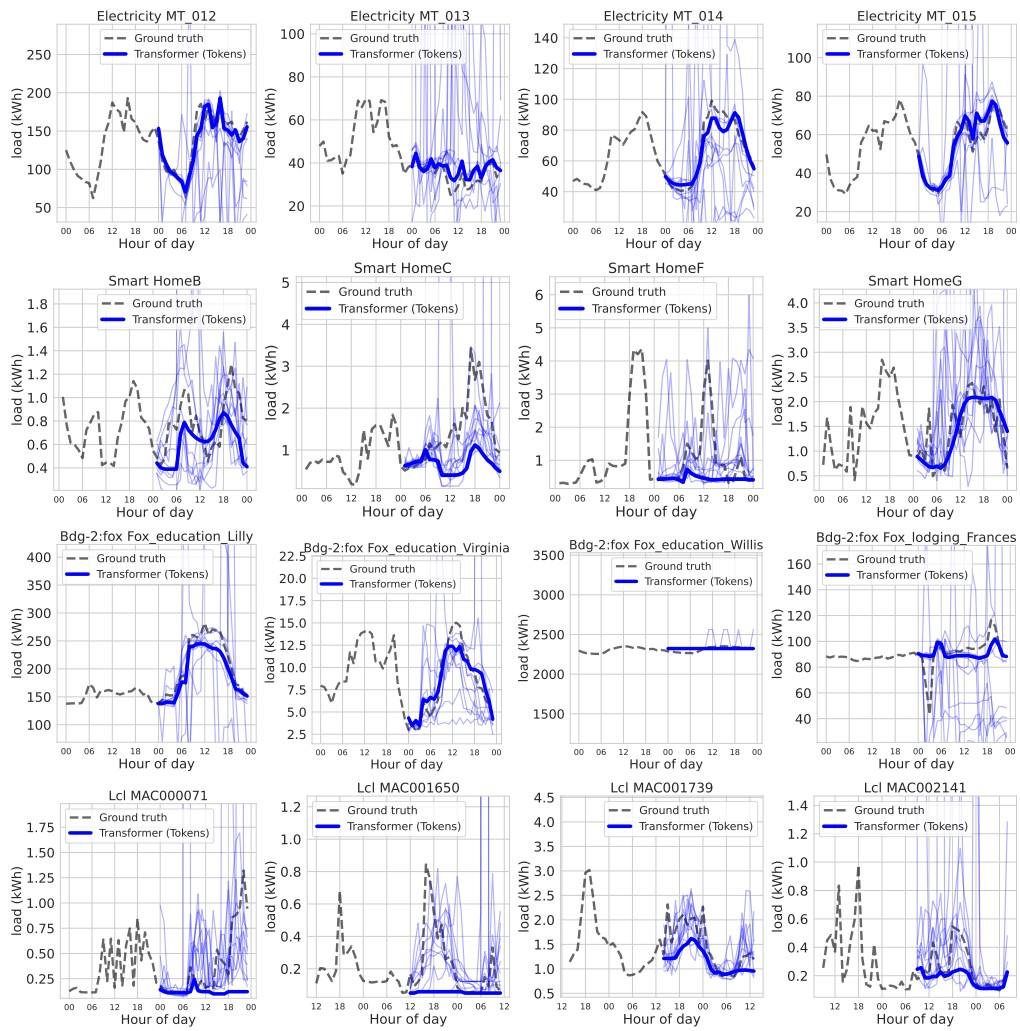

Figure 11: **Qualitative results for Transformer-L (Tokens)**. Ground truth time series are truncated to previous 24 hours for visibility. Light blue lines are 10 samples from the predicted distribution.

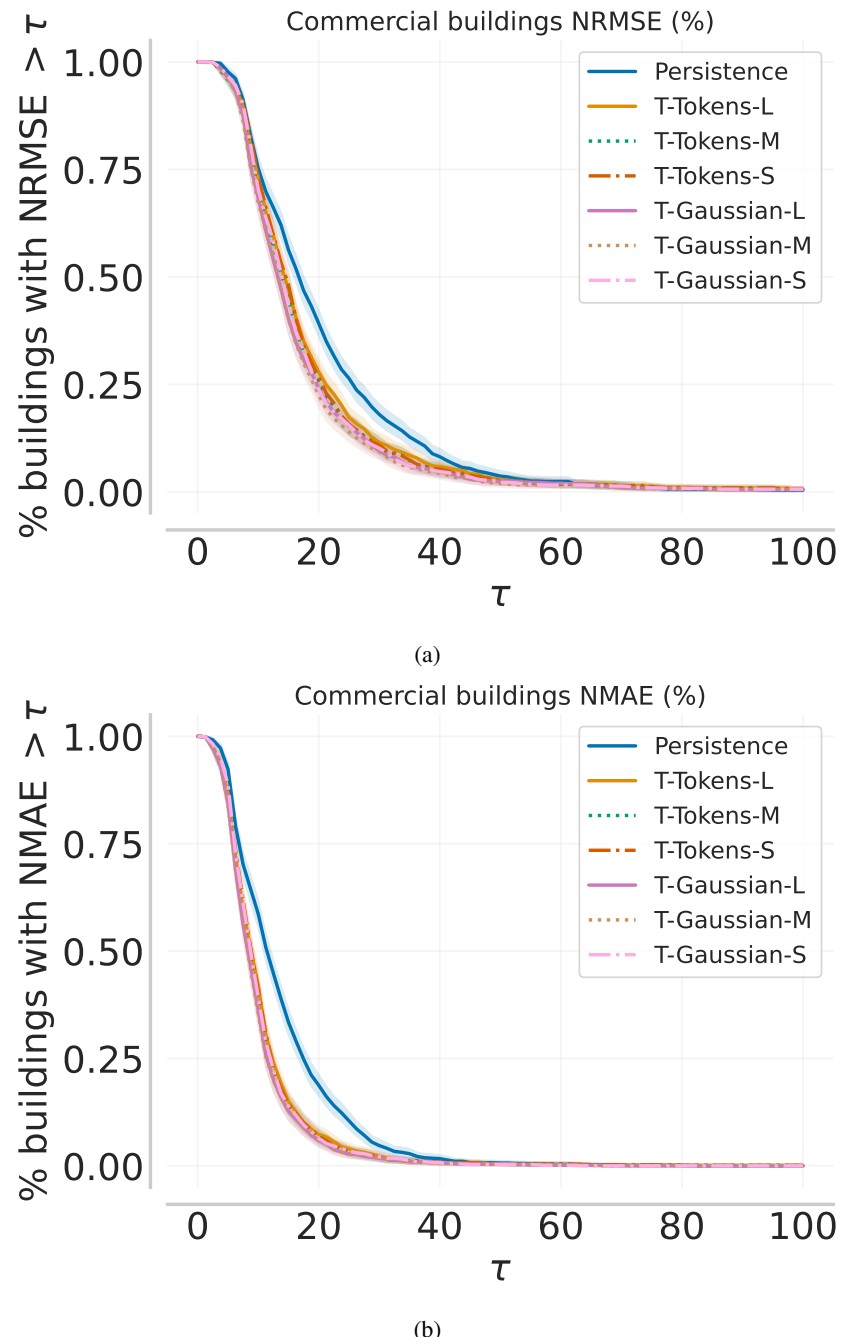

(a)

(b)

Figure 12: Real commercial building zero-shot NRMSE and NMAE performance profiles with 95% bootstrap CIs. Curves closer to the bottom left are better.

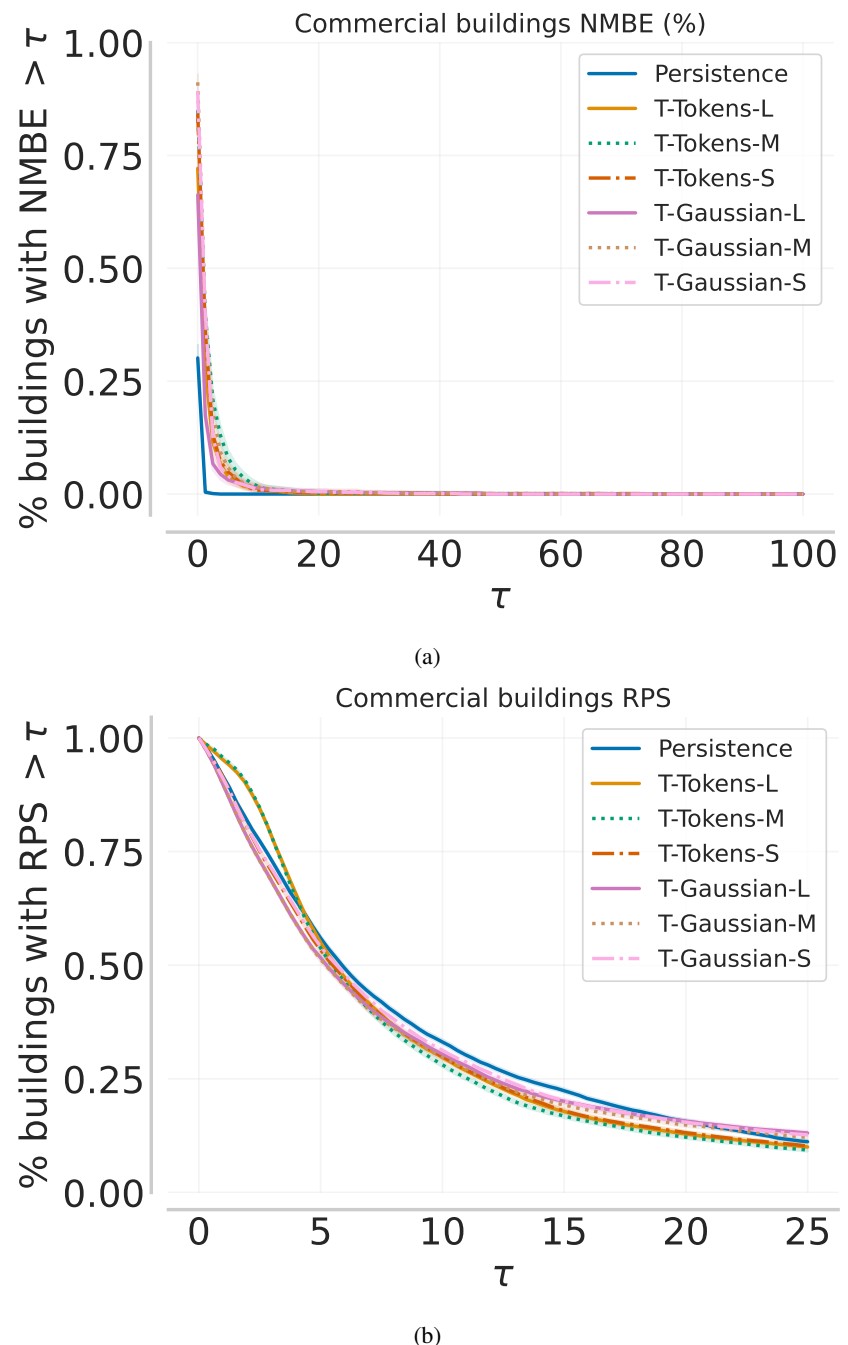

Figure 13: Real commercial building zero-shot NMBE and RPS performance profiles with 95% bootstrap CIs. Curves closer to the bottom left are better.

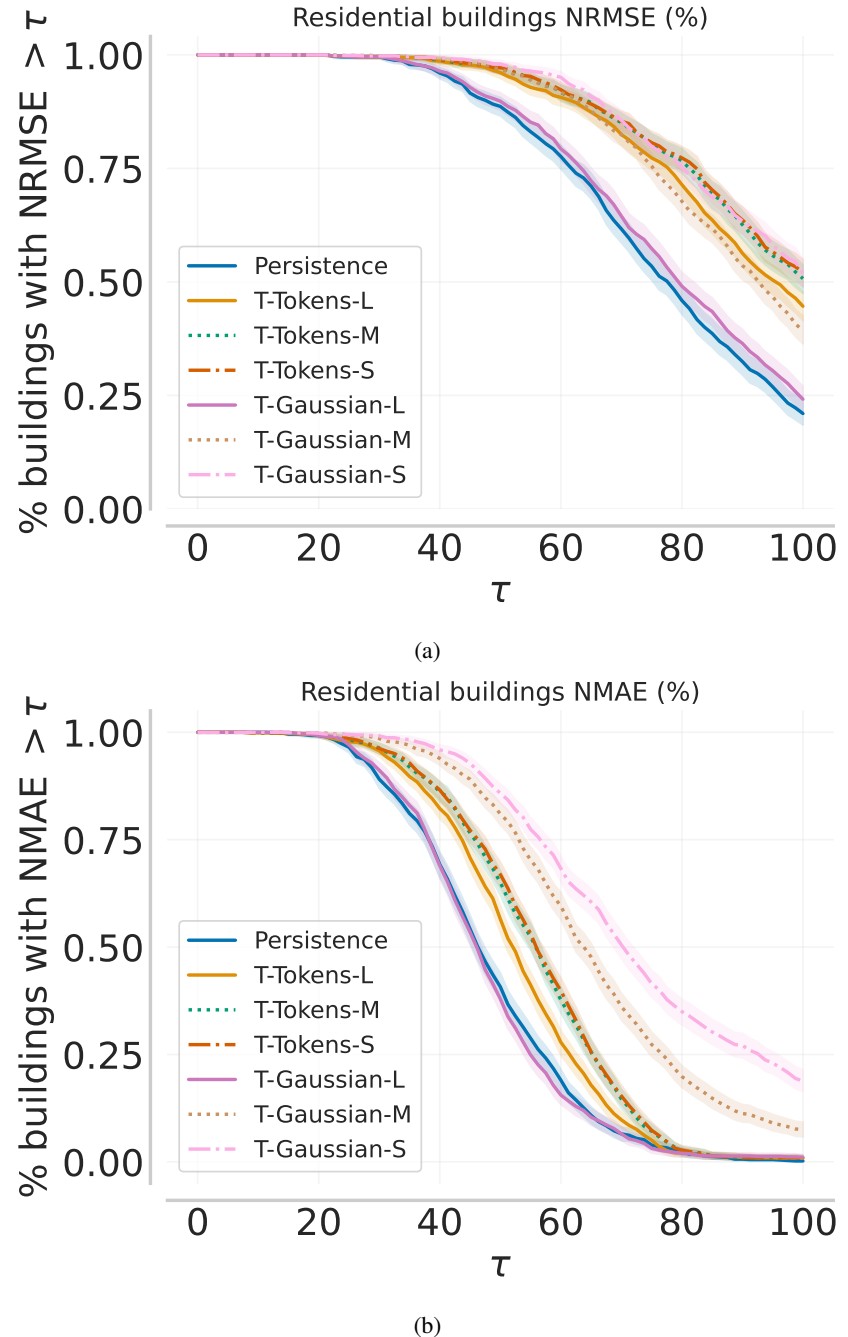

Figure 14: Real residential building zero-shot NRMSE and NMAE performance profiles with 95% bootstrap CIs. Curves closer to the bottom left are better.

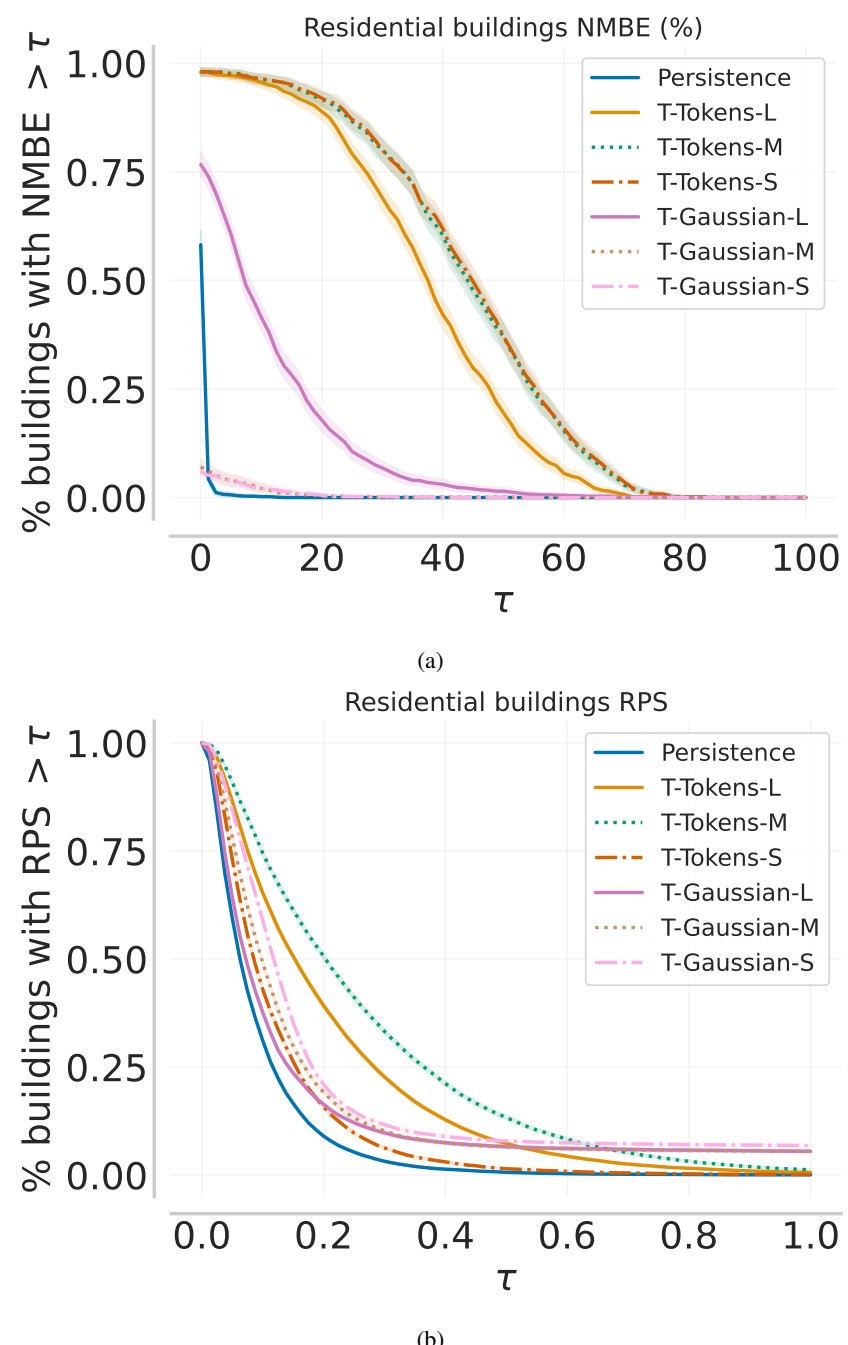

Figure 15: Real residential building zero-shot NMBE and RPS performance profiles with 95% bootstrap CIs. Curves closer to the bottom left are better.

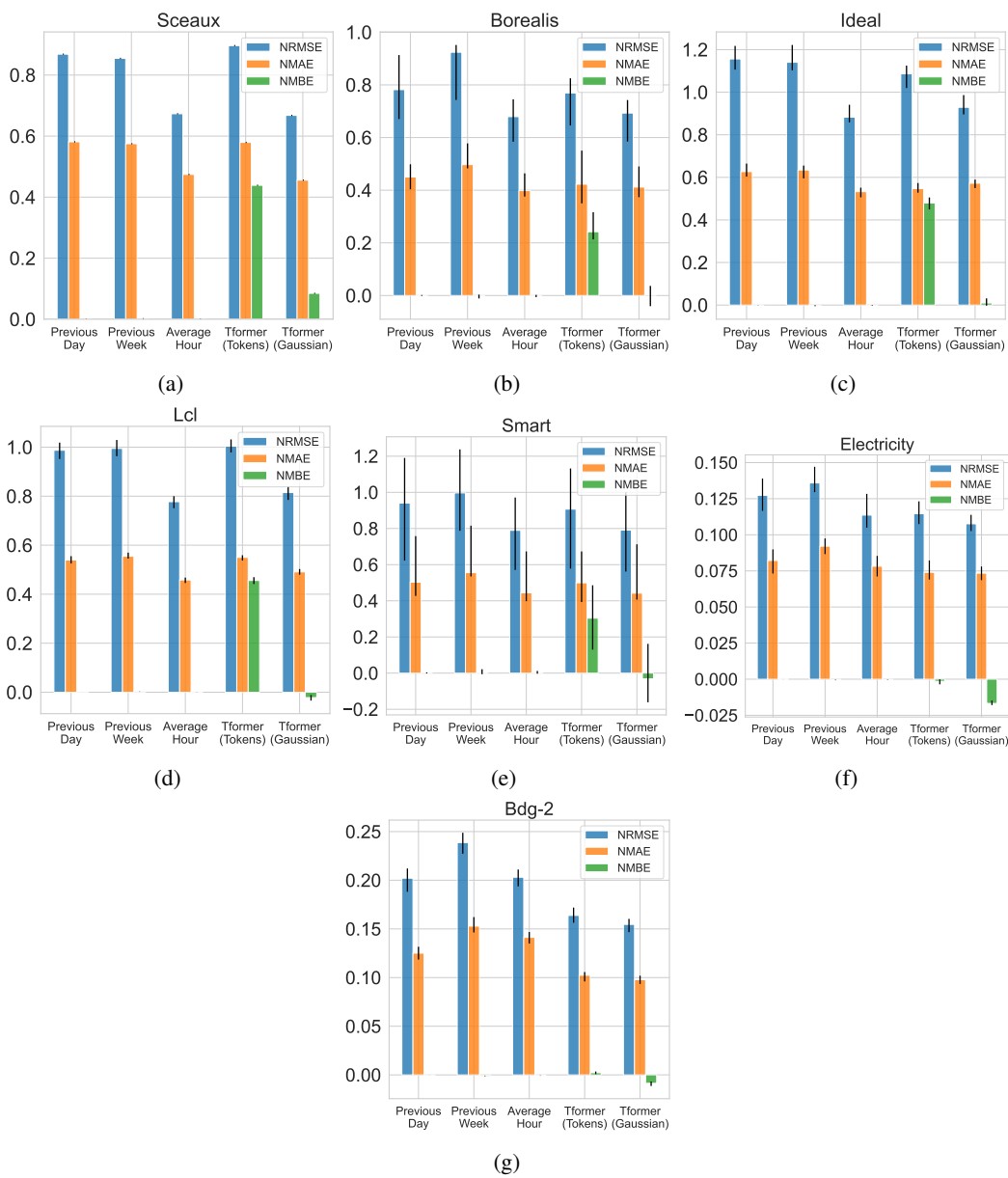

Figure 16: Per-dataset median zero-shot accuracy (NRMSE, NMAE, NMBE) with 95% bootstrap CIs. Lower is better.

