# OpenReview forum: "BuildingsBench: A Large-Scale Dataset of 900K Buildings and Benchmark for Short-Term Load Forecasting"
_NeurIPS.cc/2023/Track/Datasets_and_Benchmarks — NeurIPS 2023 Datasets and Benchmarks Poster_

### Official Review · Reviewer_zkdi · 2023-07-22
**Marginally above acceptance threshold**

**Rating:** 6
**Confidence:** 4
**Correctness:** The claims made in the submission are…
**Clarity:** The paper is well written.

**Strengths:**

1. It is quite useful to have a dataset for load forecasting. The dataset is interesting.
2. The dataset is relatively large.
3. The paper is well written.


**Additional Feedback:**

NA

**Documentation:**

Sufficient detail on data collection and organization is provided.

**Limitations:**

1. There are typos in the paper. For example, "STLF remains a challenging problem as energy demand can fluctuate heavily due to a
31 variety of unobserved and exogenous factors" ends with no period.
2. The significance of this dataset against existing ones are not well explained.


**Opportunities For Improvement:**

1. There are typos in the paper. For example, "STLF remains a challenging problem as energy demand can fluctuate heavily due to a
31 variety of unobserved and exogenous factors" ends with no period.
2. The significance of this dataset against existing ones are not well explained.


**Relation To Prior Work:**

Not very clear.

**Summary And Contributions:**

The paper presents a benchmark dataset for short-term load forecasting. The dataset is interesting. The dataset is relatively large.

---

> ### Author Response · Authors · 2023-08-23
>
> Dear reviewer zkdi, thanks for your review and positive response to our paper. We have uploaded a new version of the paper and supplementary PDF with changes highlighted in red.
>
> - We have proofread and corrected typos in the paper.
>
> - To help clarify the significance of our datasets with respect to other existing datasets, we have:
>   - Revised the following sentence in the Introduction (L52-54), which now reads: "Compared to other existing datasets (Table 1, Table 2), our large-scale pretraining and evaluation data contains both simulated and real building energy consumption time series spanning a wider range of geographic locations, years, and types of both residential \textit{and} commercial buildings".
>   - We improved the caption of Table 1: "Comparing popular building energy consumption datasets to our dataset Buildings-900K. Our dataset has significantly more buildings (residential and commercial) located across a wide geographic area and spans multiple years, which enables studying large-scale pretraining for STLF."
>
>
> Please see the main comment addressed to all reviewers for a summary of other changes we made to the paper.

---

### Official Review · Reviewer_cRNE · 2023-07-22
**This paper presents a large-scale time-series dataset from both simulation and real-world data, but there are several concerns need to be fixed.**

**Rating:** 5
**Confidence:** 3
**Clarity:** The paper is well-written.

**Strengths:**

1. This paper introduces a large-scale time series dataset, including both simulation data and real residential and commercial data. The major contribution comes from the 900K simulation data for both commercial and residential buildings and a benchmark that includes zero-shot/transfer learning tasks and several state-of-the-art baselines.

2. The authors have provided the data and code links, as well as good documentation.

**Additional Feedback:**

In Line 31, there is a period mark missing after "factors".

**Correctness:**

In general, the claims are correct. Please also see my comments 1 and 3 in **Opportunities For Improvement**.

**Documentation:**

Yes, the data details are sufficient.

**Ethics:**

No.

**Limitations:**

The authors have included a paragraph on the limitations of this benchmark. In addition, please also see my concerns and questions below.

1. Page 2, Figure 1, the data from commercial buildings present clear periodical patterns, but the residential data look more "unpredictable". For the second and third rows, I saw many spikes in the time series data. How do the authors tell if these spikes are normal conditions or "outliers" in real-world residential data?

2. Also, on Page 5, in the paragraph "processing and storage", the authors mention that they fill with zeros for the missing scenarios over one week. Is it possible to use some common matrix or tensor completion methods to approximate the missing values [1,2,3]?

References:

[1] Liu, J., Musialski, P., Wonka, P., & Ye, J. (2012). Tensor completion for estimating missing values in visual data. IEEE transactions on pattern analysis and machine intelligence, 35(1), 208-220.

[2] Yu, H. F., Rao, N., & Dhillon, I. S. (2016). Temporal regularized matrix factorization for high-dimensional time series prediction. Advances in neural information processing systems, 29.

[3] Chen, X., He, Z., & Sun, L. (2019). A Bayesian tensor decomposition approach for spatiotemporal traffic data imputation. Transportation research part C: emerging technologies, 98, 73-84.

4. On page 5, for the real data, are there any noisy or outlier scenarios that need to be considered?

5. As shown in Tables 3 and 4 on Page 7, the results show the challenges of the data from real residential buildings. The authors also provide several potential directions in Section 6.2 Residential STLF Challenges. Can the authors add more descriptions of the acceptable error range for this particular scenario? Based on my understanding, the normalized errors exceeding 100% are too large.

**Opportunities For Improvement:**

1. **Data pre-processing**: The authors use simulation data for training. I think it would improve the performance of pre-trained models by training on both simulation data and real-world data, especially for residential buildings. The imperfect real-world data may be the issue that hinders the training on both simulation and real-world data. But there are some common pre-processing tools for handling imperfect time-series data. Please see my comments on **Limitations** below.

2. **Evaluation**: The evaluation metrics can be enriched. Since this dataset aims for forecasting building energy consumption, it would be good to consider some extreme statistical metrics.

3. **Baselines**: The authors test transformer architectures, LightGBM, and linear models. I think it would strengthen the paper's quality to also consider the classic recurrent neural networks. It will help the readers to gain more insights into those common time-series techniques for different datasets.

**Relation To Prior Work:**

Yes, this paper has discussed the difference from previous work.

**Summary And Contributions:**

This paper presents a large-scale dataset, BuildingsBench, including 900K simulation data and over 1900 real-world data from both residential and commercial buildings. BuildingsBench focuses on zero-shot learning and transfer learning for short-term load forecasting (STLF). The benchmark analysis finds that (1) pre-trained models are generalizable to real commercial buildings, (2) a power-law with diminishing returns is revealed on zero-shot commercial buildings, and (3) fine-tuning pre-trained models improves the performance.

---

> ### Author Response · Authors · 2023-08-23
>
> Dear reviewer cRNE, thank you for the suggested improvements, which we have done our best to incorporate into the paper. We have uploaded a new version of the paper and supplementary PDF with changes highlighted in red.
>
> ### Outliers
>
> To help remove "unpredictable" spikes, we implemented a simple algorithm to filter spikes from the real building time series. We added the following text (L154--156): "A small fraction of spikes in the time series due to noisy meter readings are classified as outliers and filtered with a non-parametric distance-based sliding window algorithm~\citep{knox1998algorithms}". More details are in App. C.8.
>
> Most residential buildings have ~0.1% of values detected as outliers and most commercial buildings have 0. We believe a strict algorithm is favorable, as we wish to avoid false positives that may remove important load spikes that are critical to accurately forecast to avoid major faults in distribution feeders.
>
>  To summarize the results on the filtered evaluation data:
>
> - On average, we saw a 0.01% improvement in commercial NRMSE across Transformer-L models
> - On average, we saw a 2.88% improvement in residential NRMSE across Transformer-L models
>
> Please see the main comment and Table 3 for updated results.
>
> ### Missing values
>
> Thanks for the suggested references. We tried Temporal Regularized Matrix Factorization to fill missing values. However, this seemed to just replace the zeros with ~random load values near zero. We believe this did not work well because these approaches need load values from other related buildings to impute the missing values, but consumption patterns may vary widely even between related buildings. Spans of time longer than a week with missing values are rare in our evaluation suite and therefore should hardly impact the results:
>
> - ~1.4% of residential buildings per dataset in the eval suite have an instance of > 1 week of missing values
> - ~1.6% of commercial buildings per dataset in the eval suite have an instance of > 1 week of missing values
>
> ### Extreme metrics
>
> We provide performance profiles for the entire distribution of forecast errors over all test buildings in App. H. This avoids the need to select a single quantile to define an extreme error.
>
> To help explain performance profiles, we revised L231--232: "Performance profiles over all buildings **(for comparing models by examining the tails of the forecast error distribution)** and...", and L949--951: "Performance profiles plot the fraction of all buildings with performance greater than a threshold $\tau$ for a range of threshold values. **These can be used to understand the tails of the error distributions across all buildings in the benchmark. For example, these plots show the fraction of buildings with extremely high forecasting errors."** We would be interested to know whether the reviewer has any other specific metrics for extreme errors in mind.
>
> ### Large residential errors
>
> Identifying acceptable error ranges for residential building STLF is difficult as it can be problem dependent. To help give context for the error ranges, we:
>
> - Improved the caption for Table 3 as follows: "Residential NRMSEs are naturally larger than commercial buildings, because the normalization factor---the building's average consumption per hour---is small. Dividing the RMSE by a small constant causes the large NRMSE. For example, Transformer-L (Gaussian) has an RMSE of 0.72 kWh and an NRMSE of 66.57\% on the Sceaux dataset."
> - Provide reference [2], which lists the RMSEs of supervised deep learning algorithms trained on the Sceaux dataset which is in the BuildingsBench suite (see Table 7 (Hourly) in [2]). For example, the supervised LSTM achieves an RMSE of 0.7173 kWh and the supervised CNN-LSTM achieves an RMSE of 0.5957 kWh. We believe a zero-shot forecast RMSE of 0.72 kWh is reasonable.
>
> We also note that after three key improvements (see main comment---we corrected load value scales in Buildings-900K, performed extra hyperparameter tuning and outlier removal), residential NRMSEs are now below 100% in Table 3.
>
> ### Baselines
>
> We have implemented an RNN and added the results to Table 4. We added this description (L218--221):
>
> ```
> RNN (Gaussian): This is an autoregressive encoder-decoder recurrent neural network inspired by~\citet{salinas_deepar_2019}. A multi-layer LSTM~\citep{hochreiter1997long} first encodes the 168 past load values. The last encoder hidden state initializes the state of a multi-layer LSTM decoder. A linear layer maps each output of the decoder to Gaussian parameters.
> ```
>
> Results:
>
> | Commercial NRMSE | Commercial RPS | Residential NRMSE | Residential RPS |
> | --- | --- | --- | --- |
> | 41.79 | 15.28 | 96.75 | 0.078 |
>
> #### References
>
> [1] Amasyali et al. "A review of data-driven building energy consumption prediction studies." Renewable and Sustainable Energy Reviews 81 (2018).
>
> [2] Kim et al. "Predicting residential energy consumption using CNN-LSTM neural networks." Energy 182 (2019).

---

> > ### Author Response · Authors · 2023-08-29
> >
> > Dear Reviewer cRNE, as the end of author/reviewer discussions is approaching, may we know if our response addresses your main concerns? If you have any further suggestions, please let us know and we will be more than happy to engage in more discussion and improvements.
> >
> > Thank you again for devoting your time to reviewing our work.

---

### Official Review · Reviewer_ZMcD · 2023-07-25
**A novel dataset and benchmark platform to address models of energy consumption in residential and commercial buildings**

**Rating:** 8
**Confidence:** 2
**Correctness:** The dataset and the benchmark are bot…

**Strengths:**

This paper addresses the need for a large dataset to develop STLF models (pretraining), with an emphasis on forecasting energy consumption on buildings unseen by the model, for which no fine tuning has been implemented (zero-shot). In addition to providing such dataset, this submission includes a detailed framework to benchmark the aforementioned zero-shot models. The authors present a solid and sound benchmark design, which utilizes a secondary dataset they have put together. This secondary dataset (BuildingsBench)—another novel contribution in this submission—provides real buildings data collected from 7 separate sources, which is also used to fine tune the models trained on simulated data.

**Additional Feedback:**

No additional feedback. I have placed by comments, suggestions, and questions spread out in the other forms.

**Clarity:**

The paper is well written and organized. I have only a few minor points that could further improve the clarity:

- It is unclear how the speech recognition application translates to the potential for pretraining in an STLF application (line 40).
- Consistency check: the labels in Figure 1 use "timestamp" and "Datetime".
 - In section 6.3, the second to last sentence is unclear, it seems a couple of words are missing (lines 299-300).

**Documentation:**

This submission is very well documented and provides all the aspects asked for.

**Ethics:**

No ethical concerns.

**Limitations:**

It would be interesting to have some discussion on how transferable this dataset (and the benchmark platform) is to other countries. The authors discuss to some extent about the possible differences when fine tuning the model (pretrained with US data) with data from other countries, but not the other way around. For example, BuildingsBench comprises data from different geographic and cultural regions (which can drive residential consumption behavior), which perhaps do not translate directly to consumption patterns in the US (which buildings-900k only has). These cultural differences might be accentuated in the Low Carbon London dataset, where individuals in select London households were motivated to reduce their consumption (e.g., use less their appliances or prioritize high energy efficiency appliances; see the project summary report: https://innovation.ukpowernetworks.co.uk/projects/low-carbon-london/). Perhaps differences as these could explain the lower accuracy obtained in the residential buildings benchmarks.

I see no potential negative societal impact.

**Opportunities For Improvement:**

One of the limitations of the dataset, which the authors have identified, is related to the differences encountered in the benchmark performances for residential and commercial buildings, where residential buildings resulted in a considerably lower performance.

The data split for the validation data set was defined as the last 2 weeks of the time series (coinciding with the last 2 weeks of the calendar year). However, it is not discussed if a seasonal bias could be introduced, or at least a deficiency in the learning process, unless the distribution of the simulated buildings consumption is agnostic to seasonal variations (e.g., heating and cooling patterns, holiday season in the end of the year). In section 3.1  it is suggested that, indeed, there are temporal features; then it seems more likely that some temporal patterns have been missed by holding out the last two weeks.

In the benchmark analysis (section 5), the authors seek to answer the question: “Can models pretrained on Buildings-900K generalize to real buildings?”. Considering the goal of generalizing to real buildings, it seems a limitation should be recognized here, upon the decision of removing real buildings which had large consumption records (> 1.3 MWh). Those buildings might be outliers, but it is know that the 2% larger consumers (in the US) consume a third of the energy (see the CBECS 2018 report). If the Buildings-900k dataset is lacking representation of such buildings, then one could argue the dataset is not fully representative of US commercial buildings. For example, the CBECS report also shows that hospitals consume ~30 kWh/sq.ft, which for a 500k sq.ft building (being on the smaller side of hospitals) can easily reach to thousands of MW. Again, this discussion should be made in light of the goal of STLF to "help match shifting energy supply with customer demand as well as aid energy markets with accurately setting prices based on forecasted supply/demand”.

Additional possible improvements can be achieved by addressing the comments in the other forms.

**Relation To Prior Work:**

The authors present a table comparing the new dataset with 4 another similar datasets, highlighting their contribution as this dataset is two orders of magnitude larger than the others. Additionally, the authors contextualize well the need for datasets like this.

**Summary And Contributions:**

This paper presents a new development over a previously existing dataset on buildings energy consumption. The new dataset presented here is tailored for short-term load forecasting (STLF), a type of forecasting of energy demand that takes advantage of individual consumption profiles, as opposed to aggregated data. The original, existing dataset was not tailored for this task. The adaptation to an STLF task is achieved by retrieving only one variable variable of the initial dataset, which is then temporally upscaled (from 15-min resolution to 1-h resolution), in addition to a geographically-aware (county-based) organization of the data files. Further, this adaptation to the STLF task is demonstrated and benchmarked with machine learning techniques, including deep learning (transformers), showcasing that the new dataset is also applicable to approaches that are data-demanding. The benchmarks are extensive, well thought, and very well detailed. The manuscript text is well written and it is accompanied by detailed supplemental materials useful for readers looking to learn more about the methods employed.

---

> ### Author Response · Authors · 2023-08-23
>
> Dear reviewer ZMcD, thank you for your thoughtful review and positive comments. Your insights have helped us significantly improve the paper. We have uploaded a new version of the paper and supplementary PDF with changes highlighted in red. We also copy the relevant changed passages below that address your comments.
>
> ### Buildings-900K train/val split concern
>
> A question is raised about whether a seasonal bias is introduced by leaving out the last two weeks of the calendar year in our training set. We can clarify why this is not the case. There are *two* dinstinct years in the Buildings-900K training dataset, the "annual meteorological year" (AMY) 2018 and a "typical meteorological year" (TMY), a synthetic weather-year created by aggregating months from many different years. Our validation set contains the last two weeks of AMY 2018 for all buildings. Thus, our pretrained models *do* see the last two weeks of the calendar year, albeit only for the TMY weather-year. This prevents the models from failing to learn any patterns at all that only emerge during the holiday season at the end of the year.
>
> ### Is Buildings-900K fully representative of US Commercial Buildings?
>
> Thanks for raising this interesting discussion point. As indicated correctly by the reviewer, large buildings such as hospitals often have consumption values larger than 1.3 MWh. After consulting [the metadata file for commercial buildings in AMY 2018]((https://data.openei.org/s3_viewer?bucket=oedi-data-lake&prefix=nrel-pds-building-stock%2Fend-use-load-profiles-for-us-building-stock%2F2021%2Fcomstock_amy2018_release_1%2Fmetadata%2F)) provided by the NREL End-Use Load Profiles database and which lists the attributes used for the Buildings-900K simulated buildings, we confirmed that there are buildings in Buildings-900K with types such as "Hospital" and "LargeOffice" whose square footage exceeds 500k and whose consumption values exceed 1.3 MWh (> 5+ MWh). Hence, we believe that Buildings-900K is indeed representative of the U.S. commercial building stock.
>
> This investigation lead us to discover an error in how we upscaled the 15-minute consumption values from the NREL End-Use Load Profiles database to hourly values. We had averaged the 15-minute values to obtain hourly values, but this is incorrect for EnergyPlus simulations. The correct procedure is to sum the 15-minute values at four consecutive timestamps: XX:15, XX:30, XX:45, and XX+1:00 (e.g., 12:15, 12:30, 12:45, 13:00). This explains why the max hourly consumption values were small (<= 1.3 MWh).
>
> - We have re-created the Buildings-900K dataset with summed load values and will upload a new version (v1.1.0) to our OEDI data lake.
> - The correct aggregation steps (described above) are now in the revised supplementary material (App. B.4).
> - We have revised this sentence in the main text (L152-153): "We also provide an option to exclude buildings with a max hourly consumption **$>$ 5.1 MW (only 15 from Electricity)** to keep the range of consumption values similar between Buildings-900K and BuildingsBench test data, as extrapolation is not our focus".
>
> We have also retrained all models and updated the tables and figures in the paper and supplementary PDF. At the suggestion of reviewer YiaS, we conducted a (light) hyperparameter sweep to tune the learning rate and batch size of the transformers. **The results were largely unchanged after retraining on the rescaled load values with tuned hyperparameters.** To summarize, for the transformers on BuildingsBench (real):
>
> - On average, we saw a 0.44% improvement in commercial NRMSE
> - On average, we saw a 2.57% improvement in residential NRMSE
>
> Also, at the suggestion of reviewer cRNE, we implemented a simple outlier removal scheme to remove some of the spikes from the real building time series.
>
> - On average, we saw a further 0.01% improvement in commercial NRMSE
> - On average, we saw a further 2.88% improvement in residential NRMSE
>
> ### Dataset transferability limitation
>
> Thank you for bringing up this point. We agree that cultural differences may diminish the ability of a predominantly U.S.-based model to generalize to other regions. We have added the following sentences to Sec 6.3 (320--325):
>
> ```
> "Third, the stochastic occupancy model used to simulate residential consumption behavior for the pretraining data is more predictable and less chaotic than real behavior, which increases the sim-to-real gap for residential buildings. Moreover, the occupancy model does not capture behavior patterns that may appear prominently in a specific geographic region (particularly outside of the U.S.), which may hurt the generalization capabilities when deploying the model in these locations".
> ```
> We believe future improvements to residential occupancy models and the mixing of real and synthetic data in the pretraining dataset may help address this limitation.
>
> Please see the main comment for a summary of other changes we made to the paper.

---

> > ### Comment · Reviewer_ZMcD · 2023-08-31
> >
> > I am very satisfied with the detailed responses the authors have provided to my queries and comments. I continue to support the acceptance of this work.

---

> > > ### Author Response · Authors · 2023-08-31
> > >
> > > Dear Reviewer ZMcD, thank you for your positive view and continued support of our work! As it seems our responses have been satisfactory, we kindly suggest considering whether to increase the confidence score of your review. Thanks again for taking the time to review our paper.

---

### Official Review · Reviewer_YiaS · 2023-07-26
**Reviews**

**Rating:** 5
**Confidence:** 4
**Correctness:** Yes
**Clarity:** Yes

**Strengths:**

BuildingsBench provides a large-scale dataset of 900K buildings and an evaluation platform with over 1,900 real residential and commercial buildings from 7 open datasets, which can help researchers explore the pretrain-then-finetune paradigm for short-term load forecasting (STLF).

The authors show that synthetically pretrained models generalize surprisingly well to real commercial buildings, and that fine-tuning pretrained models on real commercial and residential buildings improves performance for a majority of target buildings.

The authors explore two under-explored tasks in STLF: zero-shot STLF, where a pretrained model is evaluated on unseen buildings without fine-tuning, and transfer learning, where a pretrained model is fine-tuned on a target building. This can help researchers better understand the generalizability of STLF models.

**Additional Feedback:**

The paper has several potential limitations that could be addressed to improve its quality. Firstly, a more detailed description of the methods used to generate the simulated buildings representing the U.S building stock could help readers better understand the assumptions and limitations of the dataset. Secondly, while the paper provides detailed information on the dataset, evaluation platform, and experimental results, a more detailed discussion of the limitations of the study and potential future directions for research could help readers better understand the scope and implications of the work. Finally, a more detailed analysis of the factors that contribute to the performance of different models and methods on different buildings and tasks, such as building characteristics and hyperparameters, could provide valuable insights for improving model performance and understanding the factors affecting it. By addressing these limitations, the paper could enhance its contribution to the field of building energy modeling and facilitate the adoption of the proposed approach in practical settings.

**Documentation:**

Yes

**Limitations:**

Yes

**Opportunities For Improvement:**

The paper could benefit from a more detailed description of the methods used to generate the simulated buildings representing the U.S building stock. This could help readers better understand the assumptions and limitations of the dataset.

While the paper provides detailed information on the dataset, evaluation platform, and experimental results, it could benefit from a more detailed discussion of the limitations of the study and potential future directions for research. This could help readers better understand the scope and implications of the work.

The paper could benefit from a more detailed analysis of the factors that contribute to the performance of different models and methods on different buildings and tasks. For example, the authors could explore the effect of building characteristics (e.g., size, occupancy, location) on model performance, or the effect of different hyperparameters on model performance.

**Relation To Prior Work:**

Yes

**Summary And Contributions:**

This work presents BuildingsBench, a large-scale dataset of 900K buildings and benchmark for short-term load forecasting (STLF). The dataset includes simulated buildings representing the U.S building stock and an evaluation platform with over 1,900 real residential and commercial buildings from 7 open datasets. The authors explore the pretrain-then-finetune paradigm for STLF and show that synthetically pretrained models generalize surprisingly well to real commercial buildings. They also find that increasing dataset size and diversity has diminishing returns on zero-shot commercial building performance. The paper provides detailed information on the dataset, evaluation platform, and experimental results, as well as code, data, and instructions needed to reproduce the main experimental results.

---

> ### Author Response · Authors · 2023-08-23
>
> Dear Reviewer YiaS, thank you for the suggested improvements, which we have incorporated into the paper. We have uploaded a new version of the paper and supplementary PDF with changes highlighted in red.
>
> ### Improving description of how simulated buildings were generated
>
> We have added a new paragraph on this to our datasheet in the supplementary material (App. B.3). Due to character limits we could not copy it here.
>
> ### More discussion on limitations and potential future directions
>
> As also suggested by reviewer ZMcD, we added a discussion on the generality of our pretraining dataset (L320--325):
>
> ```
> "Third, the stochastic occupancy model used to simulate residential consumption behavior for the pretraining data is more predictable and less chaotic than real behavior, which increases the sim-to-real gap for residential buildings. Moreover, the occupancy model does not capture behavior patterns that may appear prominently in a specific geographic region (particularly outside of the U.S.), which may hurt the generalization capabilities when deploying the model in these locations".
> ```
>
> Also at the request of reviewer aV5A, we discuss current limitations/future work on benchmarking SOTA transformers (L325--327):
>
> ```
> Finally, due to limited time, we only pretrained vanilla transformers on Buildings-900K.
> We encourage future work that compares these results with state-of-the-art transformers~\citep{Yuqietal-2023-PatchTST,wu2023timesnet}.
> ```
>
> We discuss plans to extend the dataset by adding auxiliary inputs including building metadata (L338--341):
>
> ```
> Other promising directions for future work include exploring joint forecasting of weather and load and the impact of building metadata on performance. To facilitate this work, we plan to update the datasets with this auxiliary information.
> ```
>
> ### Hyperparameter explorations
>
> Thanks for the suggestion to explore different hyperparameters. We have done so, to the extent possible given our limited time. The details of our hyperparameter exploration are added to App. F in the supplementary material, with the key points copied below (L915--922):
>
> ```
> We perform a grid search over the max learning rate \{6e-4, 6e-5, 6e-6\} and the batch size \{64, 128, 256\} for the two Transformer-S models. For the Transformer-S (Gaussian) model, the highest learning rate and smallest batch size had the best validation loss. However, when applying these hyperparameters to the larger models, the high learning rate of 6e-4 caused training instability. Thus, we used the lower 6e-5 learning rate, which performed best with the smallest batch size of 64 (650K total gradient updates). Similarly for the Transformer-S (Tokens) model, the highest learning rate achieved the best validation loss, but for the larger models we used the lower 6e-5 learning rate with batch size of 64.
> ```
>
> As discussed in the comment to Reviewer ZMcD, we identified a minor issue with how we aggregated 15-min to hourly load values in the pretraining dataset and recreated the dataset with correctly scaled load values. We have retrained the models on this data with tuned hyperparameters and updated the tables and figures in the paper and supplementary PDF.
>
> - On average, we saw a 0.44% improvement in commercial NRMSE
> - On average, we saw a 2.57% improvement in residential NRMSE
>
> Then, at the suggestion of reviewer cRNE, we implemented a simple outlier removal scheme to remove some of the spikes from the real building time series.
>
> - On average, we saw a further 0.01% improvement in commercial NRMSE
> - On average, we saw a further 2.88% improvement in residential NRMSE
>
> For the transfer learning experiment, our original paper explored whether to fine-tune all layers or only the last layer of the transformer. We now also conduct a learning rate sweep for the supervised ML baselines to tune their performance (L917--919):
>
> ```
> We tune the learning rate for the Linear, DLinear, and RNN baselines by sweeping over 5 values: \{1e-2, 1e-3, 1e-4, 1e-5, 1e-6\}. All non-pretrained models are trained for a maximum of 100 epochs.
> ```
>
> Summary of results:
>
> - Linear regression NRMSE improves by 17.2% on average with tuned lr
> - DLinear regression NRMSE improves by 13.2% on average after tuned lr
>
> ### More analysis of factors contributing to performance
>
> We have also added a new Fig. 4 which compares transformer model sizes vs. transfer learning performance (see also L255--261). We added this text (L285--288):
>
> ```
> The Transformer-M models demonstrate the largest performance gains due to fine-tuning, which confirms the high average probability of improvement scores (Sec.~\ref{sec:tl}). The smaller impact of fine-tuning on Transformer-L performance suggests its zero-shot performance may be nearly saturated.
> ```
>
> Please see the main comment addressed to all reviewers for a summary of other changes we made to the paper.

---

> > ### Author Response · Authors · 2023-08-29
> >
> > Dear Reviewer YiaS, as the end of author/reviewer discussions is approaching, may we know if our response addresses your main concerns? If you have any further suggestions, please let us know and we will be more than happy to engage in more discussion and improvements.
> >
> > Thank you again for devoting your time to reviewing our work.

---

### Official Review · Reviewer_aV5A · 2023-07-26

**Rating:** 7
**Confidence:** 4
**Correctness:** Yes.
**Clarity:** Yes.

**Strengths:**

1. The paper is well-organized and easy to follow.
2. The proposed dataset comprises nearly 1M time series, which can greatly help the research in building large/foundation models for time series data, or for STLF tasks.
3. The code is released and formulated well. People interested in STLF can utilize this tool to have a good start.

**Additional Feedback:**

No.

**Documentation:**

Yes.

**Ethics:**

No.

**Limitations:**

Yes, the authors have proposed solutions for the mentioned limitations.

**Opportunities For Improvement:**

1. Some typos are observed, a further proofreading may need.
2. It seems that the selected baselines are not the SOTA models in the research field of general time series forecasting. The authors may consider incorporating them in the future work. Just name a few here, PatchTST, TimesNet, etc.

**Relation To Prior Work:**

Yes.

**Summary And Contributions:**

This paper introduces BuildingsBench, an open source platform for large-scale pretraining and for benchmarking STLF problems in low-data regimes. The BuildingsBench consists of a large-scale dataset of 900K simulated buildings, and an evaluation platform with over 1.9K real buildings from 7 open datasets. Extensive STLF experiments are conducted in zero-short and few-short settings. The results show that pretraining on simulated dataset can work well on real data, and fine-tune the models can further improve the performance.

---

> ### Author Response · Authors · 2023-08-23
> **Response to Reviewer aV5A**
>
> Dear reviewer aV5A, thanks for your review and positive response to our paper. We have uploaded a new version of the paper and supplementary PDF with changes highlighted in red.
>
> - We have proofread and corrected typos in the paper.
>
> - We added a few sentences on current limitations/future work on benchmarking SOTA transformers, with citations for PatchTST and TimesNet (L325--327), copied here: "Finally, due to limited time, we only pretrained vanilla transformers on Buildings-900K. We encourage future work that compares these results with state-of-the-art transformers~\citep{Yuqietal-2023-PatchTST,wu2023timesnet}."
>
>
> Please see the main comment addressed to all reviewers for a summary of other changes we made to the paper.

---

> > ### Comment · Reviewer_aV5A · 2023-08-24
> >
> > Thank you very much for the efforts on preparing the detailed responses. I think this is a good paper, and it will be valuable to the community. I updated my confidence score.

---

> > > ### Author Response · Authors · 2023-08-25
> > >
> > > Thank you for updating your review! We are glad to hear that you find our work valuable to the community.

---

### Author Response · Authors · 2023-06-27
**Dataset DOI is now available**

Dear all,

The upload of the datasets accompanying this paper to the Open Energy Data Initiative website has been finalized (https://data.openei.org/submissions/5859)  and the data has been assigned the following DOI: https://doi.org/10.25984/1986147.

---

### Author Response · Authors · 2023-08-23
**Main response**

Dear reviewers, thank you for your suggested improvements. We are pleased by the overall positive response we received. We have replied separately to each reviewer with details on how we addressed their comments.

We have uploaded a new version of the paper and supplementary PDF with changes highlighted in red. Below, we summarize the changes made in response to reviewer comments.

### Summary of main changes

- The Buildings-900K pretraining dataset has been updated to fix a minor error in how we upscaled the 15-minute consumption values from the NREL End-Use Load Profiles database to hourly values. We needed to sum instead of average the values. This was discovered in response to a query from reviewer ZMcD about  the presence of buildings with large load values in our dataset (we verified that our dataset has buildings with > 5 MWh consumption values). This had a small impact on model performances after retraining, see below.
- We implemented a simple outlier removal scheme to remove a small fraction of the spikes from the real building time series. This was suggested by reviewer cRNE. We have created a filtered version of the evaluation datasets. This had a minor impact on model performances after retraining, see below.

We will upload the improved pretraining and evaluation datasets under BuildingsBench v1.1.0 in our OEDI data lake.
We have retrained the models and updated the results and figures in the paper for v1.1.0. We are still retraining models for the ablation studies in App. K and will update the supplementary PDF when they are ready.

Additionally:

- At the suggestion of reviewer YiaS, we conducted and added discussions about hyperparameter grid searches on the transformer learning rate and batch size, as well as learning rates of supervised baselines (App. F).
- Also at the suggestion of reviewer YiaS, we additionally improved our analysis of the factors affecting performance by presenting and discussing results on *all* transformer model sizes on transfer learning (Fig. 4). We computed probability of improvement due to fine-tuning ($P(X<Y)$) for Transformer-M in addition to Transformer-L and moved these results into the main text to improve the clarity of Table 4.
- We added an RNN baseline in response to reviewer cRNE to Table 4.
- We expanded our discussions on limitations and future work as requested by multiple reviewers (Sec 6.3, Sec 7).
- We addressed grammatical issues.
- The vocabulary size used by the Transformer (Tokens) model for v1.1.0 increased as the tokenization algorithm is sensitive to the range of consumption values and is stochastic (App. E, Fig. 6 has been updated).

### Summary of updated benchmark results

#### Zero-shot generalization (Table 3)

After fixing the scale of the load values in the pretraining dataset and tuning hyperparameters:

- On average, the Transformer-L models saw a 0.44% improvement in commercial NRMSE.
- On average, the Transformer-L models saw a 2.57% improvement in residential NRMSE.

After adding outlier removal:

- On average, we saw a further 0.01% improvement in commercial NRMSE.
- On average, we saw a further 2.88% improvement in residential NRMSE.

Trends and conclusions from the original paper largely remain unchanged. We note that after these improvements, all models in Table 3 now have lower than 100% NRMSE on the residential buildings (reviewer cRNE had concerns about large residential errors).

#### Transfer learning (Table 4, Fig. 4)

- After dataset improvements, the pretrained + fine-tuned Transformer-L (Gaussian) model is now the best model for both commercial and residential buildings on the transfer learning task (Table 4), slightly outperforming the Persistence Ensemble baseline on residential buildings (77.20 vs. 78.54 NRMSE, 0.057 vs. 0.057 RPS).
- The Transformer-M models have the highest $P(X < Y)$, with the Gaussian model achieving 98\% and 73\% respectively for commercial and residential buildings and the Tokens model scoring 70\% and 62.5\%.
For Transformer-L (Gaussian), this drops to 71\% and 68\% and to 61.5\% and 39\% for Transformer-L (Tokens) (Sec 5.3).
- Fig. 4 plots model size vs. transfer learning for all transformer model sizes, showing that the Transformer-M models benefit the most from transfer learning.

---

> ### Author Response · Authors · 2023-08-27
>
> Dear reviewers,
> we have finished re-running the ablation studies and updated the supplementary PDF (App. K). The ablated models are also pretrained on the Buildings-900K dataset with properly scaled load values and evaluated on the BuildingsBench datasets with outliers removed.
>
> Thank you again for taking the time to provide us with helpful suggestions that greatly improved our paper. We would be glad to know if the improvements we made are satisfactory.

---

### Decision · Program_Chairs · 2023-09-22

**Decision:**

Accept (Poster)

**Comment:**

Based on the feedback from the reviewers and the responses from the authors, the paper titled "BuildingsBench" has received overall positive reviews. The paper introduces a substantial dataset for short-term load forecasting and provides valuable contributions to the field. The strengths of the paper include its organization, the size and quality of the dataset, and the clarity of the writing.

However, there were some common areas for improvement highlighted by the reviewers. These include addressing minor typos in the paper, providing a more comprehensive explanation of the dataset's significance compared to existing ones, and improving the clarity of the paper's relation to prior work.

The authors have responded to the reviewers' comments diligently and made appropriate revisions to the paper. They have corrected typos, clarified the dataset's significance, and improved the paper's relation to prior work. Additionally, they have made substantial efforts to address more specific concerns raised by reviewers, such as data preprocessing, evaluation metrics, baselines, data spikes, and missing values. The authors' responses have generally been thorough and effective.

Overall, the paper has been significantly improved based on the reviewers' feedback, and the authors have adequately addressed the concerns raised.